# TOLNet validation of satellite ozone profiles in the troposphere: impact of retrieval wavelengths

Matthew S. Johnson[1], Alexei Rozanov[2], Mark Weber[2], Nora Mettig[2], John Sullivan[3], Michael J. Newchurch[4], Shi Kuang[5], Thierry Leblanc[6], Fernando Chouza[6], Timothy A. Berkoff[7], Guillaume Gronoff[7,8], Kevin B. Strawbridge[9], Raul J. Alvarez[10], Andrew O. Langford[10], Christoph J. Senff[11], Guillaume Kirgis[11,13], Brandi McCarty[11], Larry Twigg[12]

[1]Earth Science Division, NASA Ames Research Center, Moffett Field, CA, USA
[2]University of Bremen, Institute of Environmental Physics, Bremen, Germany
[3]NASA Goddard Space Flight Center, Greenbelt, MD, USA
[4]Atmospheric Science Department, University of Alabama in Huntsville, Huntsville, AL, USA
[5]Earth System Science Center, University of Alabama in Huntsville, Huntsville, AL, USA
[6]Laboratory Studies and Atmospheric Observations, Jet Propulsion Laboratory,
California Institute of Technology, Wrightwood, CA, USA
[7]NASA Langley Research Center, Hampton, VA, USA
[8]Science Systems and Applications Inc, Hampton, VA, USA
[9]Air Quality Processes Research Section, Environment and Climate Change Canada, Toronto, ON, Canada
[10]NOAA Earth System Research Laboratory (ESRL) Chemical Sciences Division, Boulder, CO, USA
[11]University of Colorado Cooperative Institute for Research in Environmental Sciences (CIRES) at the NOAA Earth System Research Laboratory (ESRL) Chemical Sciences Division, Boulder, CO, USA
[12]Science Systems and Applications Inc, Lanham, MD, USA
[13]now at: 2210 Kirby Ave, Chattanooga, TN, USA

*Correspondence to*: Matthew S. Johnson (matthew.s.johnson@nasa.gov)

**Abstract.** The Tropospheric Ozone Lidar Network (TOLNet) was applied to validate retrievals of ozone ($O_3$) profiles in the troposphere from the TROPOspheric Monitoring Instrument (TROPOMI) ultraviolet (UV), Cross-track Infrared Sounder (CrIS) infrared (IR), and a combined UV+IR wavelength retrieval from TROPOMI/CrIS. Observations from six separate ground-based lidar systems and various locations of ozonesondes distributed throughout North America and in the Netherlands were applied to quantify systematic bias and random errors for each satellite retrieval. Furthermore, TOLNet data were used to intercompare idealized UV, IR, and UV+IR convolved lidar profiles of $O_3$ in the troposphere during case studies representative of high $O_3$ events. This study shows that the improved sensitivity and vertical resolution in UV+IR retrievals in the middle- and upper-troposphere resulted in tropospheric degree of freedom (DOF) values ~33% higher compared to UV- and IR-only retrievals. The increased DOFs in the UV+IR retrievals allowed for improved reproduction of mid- and upper-tropospheric $O_3$ enhancements, and to a lesser degree near-surface pollution enhancements, compared to single wavelength satellite products.

The validation of $O_3$ profiles in the troposphere retrieved with the UV-only, IR-only, and UV+IR Tikhonov regularised Ozone Profile retrievAl with SCIATRAN (TOPAS) algorithm developed at the Institute for Environmental Physics, University of Bremen demonstrated the utility of using TOLNet as a satellite evaluation data set. TOPAS UV-only, IR-only, and UV+IR wavelength retrievals had systematic biases, quantified with normalized mean bias, throughout the troposphere of 11.2 ppb (22.1%), -1.7 ppb (-0.3%), and 3.5 ppb (7.8%), respectively, which meet the tropospheric systematic bias requirements defined by the science teams for the TROPOMI and CrIS sensors. The primary drivers of systematic bias were determined to be solar zenith angle, surface albedo, and cloud fraction.

Random errors, representative of uncertainty in the retrievals and quantified by root mean squared errors (RMSE), were large for all three retrievals with UV-only, IR-only, and UV+IR wavelength retrievals having RMSE throughout the troposphere of 17.4 ppb (19.8% of mean tropospheric column values), 10.5 ppb (12.6% of mean tropospheric column values), and 14.0 ppb (14.6% of mean tropospheric column values), respectively. TOPAS UV-only profiles did not meet the uncertainty requirements defined for TROPOMI for the troposphere; however, CrIS IR-only retrievals did meet the uncertainty requirements defined by this mission. The larger random biases reflect the challenge of retrieving daily $O_3$ profiles due to the limited sensitivity and vertical resolution of these retrievals in the troposphere. Tropospheric systematic biases and random error were lower in IR-only and combined UV+IR retrievals compared to UV-only products due to the increased sensitivity in the troposphere allowing the retrievals to deviate further from the a priori profiles. Observations from TOLNet demonstrated that the performance of the three satellite products varied by season and altitude in the troposphere. TOLNet was shown to result in similar validation statistics compared to ozonesonde data, which are a commonly-used satellite evaluation data source, demonstrating that TOLNet is a sufficient source of satellite $O_3$ profile validation data in the troposphere which is critical as this data source is the primary product identified for the tropospheric $O_3$ validation of the recently-launched Tropospheric Emissions: Monitoring of Pollution (TEMPO) mission.

## 1 Introduction

Consistent observations of tropospheric ozone ($O_3$) are critical for understanding atmospheric chemistry, important societal issues such as air quality and human health (WHO, 2003; US EPA, 2006), and long-term trends in atmospheric chemical composition (Cooper et al., 2014). Monitoring tropospheric $O_3$ is typically done with ground-based in situ measurement networks, tropospheric $O_3$ lidars, and ozonesonde launches (Lefohn et al., 2018; Tarasick et al., 2019; Sullivan et al., 2022). These observation types provide high accuracy information; however, surface-level monitoring networks do not detect $O_3$ vertical profiles throughout the tropospheric column and ozonesondes and lidars are spatiotemporally sparse. To fill this time and space void, over the past couple decades satellite sensors have been developed to retrieve $O_3$ profiles in the stratosphere and troposphere with near global coverage (Hoogen et al., 1999; Liu et al., 2005). However, due to the coarse vertical resolution of nadir-viewing passive satellite retrievals of $O_3$ profiles in the troposphere (>6 km) the representativeness and accuracy of this data source can be degraded compared to ozonesondes and lidars. Given the benefit from the observational coverage of satellites, it is vital to quantify these sensor's systematic biases and unresolved errors in the troposphere.

Vertical profiles of $O_3$ in the troposphere have been retrieved by satellites for multiple decades and Table 1 summarizes some of the most commonly used spaceborne sensors. The first spaceborne sensor to retrieve tropospheric $O_3$ vertical profiles was the Global Ozone Monitoring Experiment (GOME) instrument which was launched in 1995 onboard the European Space Agency (ESA) European Remote Sensing Satellite (ERS-2) (Burrows et al., 1999). This ultraviolet (UV) wavelength (between 237–406 nm) retrieval (Hoogen et al., 1999; Liu et al., 2005) from GOME had a vertical resolution of 10–15 km in the troposphere and spatial resolution of 40 km × 320 km (Liu et al., 2005). A follow-on sensor for continued vertical profiling of tropospheric $O_3$, GOME-2, was launched in 2006 onboard the ESA MetOp-A satellite (Callies et al., 2000). GOME-2 applies an UV wavelength (between 240–403 nm) retrieval

and has a ground pixel size of 40 km × 80 km with vertical resolution of 7–15 km in the troposphere (Miles et al., 2015; Kauppi et al., 2016). National Aeronautics and Space Administration (NASA) launched the polar-orbiting Aura satellite in 2004 which is the platform for the Dutch-Finnish nadir viewing spectrometer Ozone Monitoring Instrument

(OMI) currently still retrieving tropospheric $O_3$ profiles (Liu et al., 2010). There are three $O_3$ profile retrieval algorithms for OMI (NASA - Royal Netherlands Meteorological Institute (KNMI), van Oss et al., 2002; Smithsonian Astrophysical Observatory (SAO), Liu et al., 2010; Rutherford Appleton Laboratory (RAL) Space, Pope et al., 2023). The SAO algorithm uses UV wavelengths (270–330 nm) to provide data at a spatial resolution of 13 km × 48 km and vertical resolution in the troposphere of 10–14 km (Liu et al., 2010; Bak et al., 2013). The NASA-KNMI OMI

algorithm uses the same UV wavelengths resulting in similar spatial and vertical resolution in the troposphere as the SAO product (Kroon et al., 2011). The RAL Space algorithm uses UV wavelengths (270–350 nm) to retrieve $O_3$ profiles at the native spatial resolution of the sensor (13 km × 24 km at nadir) with similar vertical resolution as the other two algorithms (Miles et al., 2015; Keppens et al., 2018; Pope et al., 2023). Finally, TROPOspheric Monitoring Instrument (TROPOMI) was launched onboard the ESA's Sentinel-5 Precursor (S5P) satellite in 2017 and retrieves

tropospheric $O_3$ profiles with relatively high spatial resolution (28.8 km × 5.6 km) and vertical resolution of 10-15 km in the troposphere using UV wavelengths (270–330 nm) (Mettig et al., 2021).

**Table 1. Information about some of the recent UV and IR satellite sensors retrieving $O_3$ vertical profiles in the troposphere.**

| Sensor | Years Active | Wavelengths | Horizontal Resolution (km) | Vertical Resolution (km) |
|---|---|---|---|---|
| GOME | 1995 – 2011 | UV: 237–406 nm | 40 km × 320 km | 10 – 15 |
| GOME-2 | 2006 – present | UV: 240–403 nm | 40 km × 80 km | 7 – 15 |
| OMI | 2004 – present | UV: 270–350 nm | 13 km × 48 km (SAO/KNMI) 13 km × 24 km (RAL) | 10 – 14 |
| TROPOMI | 2017 – present | UV: 270–330 nm | 28.8 km × 5.6 km | 10 – 15 |
| AIRS | 2002 – present | TIR: 985–1318 cm | 50 km × 50 km | 6 – 8 |
| TES | 2004 – 2018 | TIR: 995–1070 cm | 5 km × 8 km | 6 – 7 |
| IASI | 2006 – present | TIR: 975–1100 cm | 12 km × 25 km to 48 km × 50 km | 6 – 8 |
| CrIS | 2011 – present | TIR: 650–1095 cm | 42 km × 42 km | 4 – 10 |

Spaceborne sensors using thermal infrared (TIR) wavelengths such as the Infrared Atmospheric Sounding

Interferometer (IASI) (Clerbaux et al., 2010), Atmospheric Infrared Sounder (AIRS) (Chahine et al., 2006), Tropospheric Emission Spectrometer (TES) (Beer et al., 2001), and Cross-track Infrared Sounder (CrIS) (Ma et al., 2016) also retrieve tropospheric $O_3$ vertical profiles. Three IASI sensors have been launched to provide continuous data from 2006 (onboard MetOp-A) to present (onboard MetOp-C) using multiple algorithms applying TIR wavelengths between 975–1100 cm (Keim et al., 2009; Hurtmans et al., 2012). Tropospheric $O_3$ vertical profiles from

IASI sensors have similar spatial resolution as UV-based retrievals (from 12 km × 25 km to 48 km × 50 km) with higher vertical resolution in the troposphere (6–8 km) compared to UV-based sensors (Boynard et al., 2009). TES, also onboard NASA's Aura satellite, uses TIR wavelengths (995–1070 cm) to retrieve $O_3$ vertical profiles with high

spatial resolution (5 km × 8 km) and similar vertical resolution as IASI (6–7 km in the troposphere) (Worden et al., 2007a). The NASA Aqua satellite was launched in 2002 which is the platform for the AIRS TIR sensor which retrieves

$O_3$ profiles at ~50 km × 50 km spatial resolution with vertical resolution in the troposphere of 6–8 km using wavelengths between 985–1318 cm (Fu et al., 2018). The National Oceanic and Atmospheric Administration (NOAA) Suomi National Polar-orbiting Partnership (Suomi-NPP) satellite, which houses CrIS, was launched in 2011 and retrieves $O_3$ profiles in the TIR wavelengths (650–1095 cm) at 42 km × 42 km spatial resolution and vertical resolution between 4-10 km (Ma et al., 2016).

Given the higher vertical resolution of TIR retrievals compared to UV-only sensors in the troposphere, studies have been conducted to demonstrate the improvements in $O_3$ vertical profile retrievals when combing both wavelength ranges (e.g., Natraj et al., 2011). This has been demonstrated by combining retrievals from OMI+TES (Worden et al., 2007b; Fu et al., 2013; Colombi et al., 2021), GOME-2+IASI (Cuesta et al., 2013), and OMI+AIRS (Fu et al., 2018). Multiple recent studies have combined UV+IR wavelength retrievals from two newer satellite sensors TROPOMI and

CrIS to retrieve tropospheric $O_3$ vertical profiles (Mettig et al., 2022; Malina et al., 2022). The combined UV+IR TROPOMI/CrIS $O_3$ profile retrievals from Mettig et al. (2022) were evaluated in the troposphere for a full-year between 2018-2019 using a small sample (2 lidar systems which are also part of the Tropospheric Ozone Lidar Network (TOLNet)) of ground-based lidar remote-sensing observations from the Network for the Detection of Atmospheric Composition Change (NDACC) and ozonesondes (i.e., World Ozone and Ultraviolet Radiation Data

Center (WOUDC) and the Southern Hemisphere Additional Ozonesondes (SHADOZ)) and demonstrated that the combined UV+IR retrievals were more consistent with observations compared to the UV-only product. Malina et al. (2022) also evaluated a full-year (between 2019-2020) of combined UV+IR TROPOMI/CrIS $O_3$ profiles using correlative satellite retrievals and ozonesondes and further showed that combined UV+IR retrievals were more accurate in the troposphere compared to UV- and IR-only products. Mettig et al. (2022) and Malina et al. (2022) both

combined TROPOMI and CrIS retrievals; however, applied different retrieval algorithms, a priori input data, and portions of the spectral bands from each satellite, thus the validation results differed to some degree which is discussed in the current manuscript.

One of the primary goals of TOLNet is to validate tropospheric $O_3$ retrievals from the recently-launched NASA Tropospheric Emissions: Monitoring of Pollution (TEMPO) geostationary satellite mission (Chance et al.,

2013; Zoogman et al., 2017). Demonstrating the capability of TOLNet to sufficiently validate satellite $O_3$ profiles is vital as TOLNet is the primary validation data source for validating TEMPO $O_3$ products in the troposphere. To-date, no studies have validated satellite data with TOLNet beyond 1 or 2 individual systems instead of the entire network (8 total lidar systems) (Mettig et al., 2022; Sullivan et al., 2022). TEMPO will retrieve $O_3$ profiles, along with partial columns including lowermost tropospheric (0–2 km above ground level (agl)) values, using combined UV (290–345

135    nm) and visible (VIS, 540–650 nm) wavelengths (Natraj et al., 2011; Chance et al., 2013; Zoogman et al., 2017). UV+VIS retrievals of $O_3$ profiles have increased sensitivity to $O_3$ in the lower troposphere when compared to UV-only sensors (Natraj et al., 2011; Zoogman et al., 2017). TEMPO will provide 1-2 hour averaged tropospheric column, 0-2 km partial columns, and $O_3$ profiles at a high spatial resolution of 8.0 km × 4.5 km.

This study builds upon Mettig et al. (2022) to demonstrate the full capability of TOLNet (6 of the 8 systems
that were available for the first year of TROPOMI observations) to validate satellite $O_3$ retrievals at multiple vertical
levels in the troposphere. This study applies all available TOLNet systems with spatial coverage throughout the US
and in the Netherlands, compared to the small subset of 2 lidar systems used in Mettig et al. (2022), to conduct a more
robust validation of the UV-only TROPOMI, IR-only CrIS, and UV+TIR TROPOMI/CrIS $O_3$ profile retrievals.
Furthermore, the only other study to validate TROPOMI/CrIS UV+IR retrievals (Malina et al., 2022) did not apply
any ground-based lidar observations. Finally, this study conducts a detailed statistical analysis of satellite $O_3$ profile
retrievals at various vertical levels of the troposphere and investigates the capability of these retrievals to reproduce
anomalous atmospheric composition with large impacts on air quality (e.g., stratospheric intrusions, lowermost
troposphere pollution events) which was not conducted in past studies validating TROPOMI/CrIS UV+IR retrievals
in the troposphere (Mettig et al., 2022; Malina et al., 2022). The manuscript is organized in the following manner:
Sect. 2 describes the TOLNet data, which serves as the primary validation data set, ozonesondes, and satellite data
products applied in this study; Sect. 3 presents the results of the validation; Sect. 4 discusses the overall systematic
biases and random errors of the retrievals; and Sect. 5 includes the conclusions of the study.

## 2 Methods

### 2.1 TOLNet

TOLNet was established in 2011 and consists of 8 lidar systems distributed throughout North America (Newchurch
et al., 2016; https://tolnet.larc.nasa.gov/). Figure 1 shows the geographic locations of the home sites for each of the
lidars making up TOLNet. The primary goals of TOLNet are to provide data for: a) understanding physicochemical
processes controlling tropospheric $O_3$ concentrations and morphology, b) evaluation of satellite profile products
retrieving tropospheric $O_3$, and c) chemical transport and air quality model evaluation. TOLNet measurements provide
high vertical and temporal resolution data with minimal systematic bias (~5%) and sufficient precision between 0%
to 20% depending on specific systems, time of day, altitude, and temporal/vertical averaging (Leblanc et al., 2016,
2018). These high resolution observations with minimal bias/error are a desirable satellite validation data set and have
been used to evaluate and better understand $O_3$ profile retrievals (e.g., Johnson et al., 2018; Sullivan et al., 2022;
Mettig et al., 2021, 2022). However, to-date, the full complement of TOLNet lidars have not been used to validate
satellite $O_3$ vertical profiles.

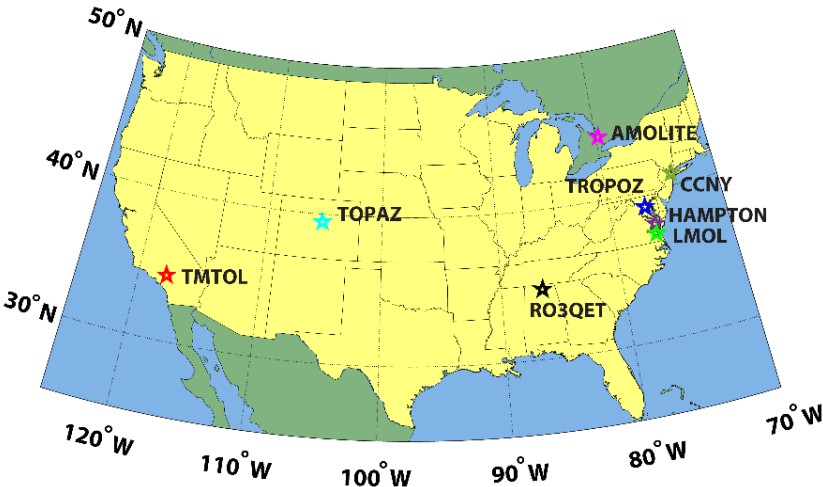

**Figure 1. Locations of home stations for the lidar systems of TOLNet (https://tolnet.larc.nasa.gov/).**

During the years just after the launch of TROPOMI, TOLNet lidar systems made dedicated observations during the overpass time of this spaceborne sensor ($\pm$ 1 hour) for validation (L2 CALVAL). For this study, there are a total of 185 TOLNet observations for validation during the time of TROPOMI/CrIS data availability between 2018-2019 (see details in Sect. 2.3). Differential Absorption Lidar (DIAL)-derived vertically resolved $O_3$ from 6 of the 8 TOLNet lidar systems were applied to validate UV-only TROPOMI, IR-only CrIS, and UV+TIR TROPOMI/CrIS $O_3$ profile retrievals. Data from the following TOLNet stations were applied: 1) NASA Jet Propulsion Laboratory (JPL) Table Mountain tropospheric ozone lidar (TMTOL) (McDermid et al., 2002), 2) NOAA Tunable Optical Profiler for Aerosol and oZone Lidar (TOPAZ) (Alvarez et al., 2011), 3) University of Alabama in Huntsville Rocket-city $O_3$ Quality Evaluation in the Troposphere lidar (RO$_3$QET) (Kuang et al., 2013), 4) Autonomous Mobile Ozone Lidar for Tropospheric Experiments (AMOLITE) (Strawbridge et al., 2018), 5) NASA Langley Mobile Ozone Lidar (LMOL) (De Young et al., 2017; Gronoff et al., 2019; Farris et al., 2019), and 6) NASA Goddard Space Flight Center mobile Tropospheric Ozone Lidar (TROPOZ) (Sullivan et al., 2014). Table 2 provides the basic information about these lidar systems used for validation during 2018-2019.

A portion of TOLNet systems are mobile (e.g., LMOL, TROPOZ, TOPAZ, AMOLITE, and RO$_3$QET) and were not located solely at their home stations between 2018-2019. For instance, during the summers of 2018 and 2019, LMOL took observations at NASA Langley Research Center, VA (LaRC), Hart Miller Island, MD (HMI), and Sherwood Island, CT (SIC). The majority of the lidar systems applied here were distributed throughout the United States while TROPOZ was in the Netherlands based at the Cabauw Experimental Site for Atmospheric Research (CESAR) for the 2019 TROpomi vaLIdation eXperiment (TROLIX-19) campaign (Sullivan et al., 2022). TOLNet provides observations ideal for satellite validation as the lidars are located at various locations which experience variable atmospheric composition and meteorological conditions and have different viewing conditions (e.g., surface reflectivity, system altitudes, topography, solar zenith (sza) and viewing angles). The lidars also take measurements in all seasons throughout the year providing a robust validation data set. This study includes 13, 28, 78, and 66 TOLNet observations for the winter (DJF), spring (MAM), summer (JJA), and fall (SON) months, respectively. Furthermore,

the higher vertical resolution of ground-based lidars, compared to satellite profile products, make these observations ideal for validating satellite $O_3$ profiles in the troposphere at various altitudes.

**Table 2. TOLNet lidar system and ozonesonde observation information between 2018-2019 used for satellite validation.**

| TOLNet | | | | |
|---|---|---|---|---|
| **System Name** | **Latitude (°N)** | **Longitude (°E)** | **Elevation (m asl)** | **Observations (number of days)** |
| TMTOL | 34.38 | -117.68 | 2285 | 64 |
| TOPAZ | 39.99 | -105.26 | 1674 | 23 |
| RO$_3$QET | 34.73 | -86.65 | 206 | 25 |
| AMOLITE | 57.18 | -111.64 | 266 | 22 |
| LMOL – LaRC | 37.09 | -76.38 | 3 | 23 |
| LMOL – HMI | 39.24 | -76.36 | 6 | 12 |
| LMOL – SIC | 41.11 | -73.34 | 3 | 6 |
| TROPOZ | 51.97 | 4.93 | 3 | 10 |

| Ozonesondes | | | | |
|---|---|---|---|---|
| **Location Name** | **Latitude (°N)** | **Longitude (°E)** | **Elevation (m asl)** | **Observations (number of days)** |
| HUB | 39.06 | -76.88 | 67 | 7 |
| WCT | 41.11 | -73.34 | 3 | 3 |
| FLP | 39.24 | -73.14 | 4 | 7 |
| HMI | 39.24 | -76.36 | 6 | 6 |
| RU | 40.47 | -74.43 | 19 | 2 |
| UMBC | 39.25 | -76.71 | 60 | 6 |
| UAH | 34.73 | -86.64 | 196 | 4 |
| GML | 39.95 | -105.20 | 1743 | 16 |

**2.2 Ozonesondes**

In addition to TOLNet data, ozonesonde observations from the same time period and similar locations were applied for validating satellite $O_3$ profiles. Ozonesonde data from launches at Howard University – Beltsville, MD (HUB), Westport, CT (WCT), Flaxpond, NY (FLP), HMI, Rutgers University, NJ (RU), University of Maryland – Baltimore Country, MD (UMBC), University of Alabama in Huntsville, AL (UAH), and the Global Modeling Laboratory (GML) in Boulder, CO were applied (see Table 2). Ozonesondes have been used extensively to validate satellite $O_3$ vertical profiles in past research (e.g., Worden et al., 2007a; Kroon et al., 2011; Verstraeten et al., 2013; Huang et al., 2017; Malina et al., 2022). In addition to the fact that TOLNet lidar data has been shown to be highly accurate (Leblanc et al., 2016, 2018) as discussed above, this study intercompares the validation statistics from spatially and temporally collocated TOLNet and ozonesonde observations to demonstrate the capability of TOLNet to be used for validating satellite $O_3$ vertical profiles. In all, we apply 51 ozonesonde observations for validation of satellite data between 2018-2019. The seasonal distribution of these ozonesondes were: 2, 2, 39, and 8 for the winter, spring, summer, and fall months, respectively. In order to have a direct comparison of the validation using ozonesonde and TOLNet, we use

ozonesondes which were nearly directly spatially and temporally co-located with lidar systems as shown in the location information provided in Table 2. Similar to TOLNet data, the ozonesondes provide high vertical resolution (effective resolution ~100 m) $O_3$ information with high accuracy (<15% below ~20 km agl) (e.g., Witte et al., 2018; Sterling et al., 2018; Thompson et al., 2019) from locations distributed throughout the United States in regions which experience variable atmospheric composition, meteorological conditions, and spaceborne sensor viewing conditions.

### 2.3 Satellite $O_3$ profile retrievals

This study validates $O_3$ profiles in the troposphere retrieved with the University of Bremen Tikhonov regularised Ozone Profile retrievAl with SCIATRAN (TOPAS) algorithm which was applied to TROPOMI UV-only, CrIS IR-only, and TROPOMI/CrIS UV+IR data (Mettig et al., 2021, 2022). Mettig et al. (2021, 2022) describe the three-step iterative TOPAS retrieval in detail which is based on the first-order Tikhonov regularization approach (Tikhonov, 1963). Briefly, retrievals of $O_3$ vertical profiles ($x_i$) from the surface to 60 km asl, provided in 1 km bins, are done employing Eq. (1):

$$x_{i+1} = x_i + [K^T S_y^{-1} K + S_r]^{-1} \times [K^T S_y^{-1}(y - F(x_i) - S_r(x_i - x_a)], \tag{1}$$

where index $i$ denotes the iteration number. Climatological a priori $O_3$ profiles ($x_a$) are applied in the retrieval and the profile shapes are determined based on total $O_3$ column abundances (Lamsal et al., 2004). A priori standard deviation is assumed to be 30% and is accounted for in the regularization matrix ($S_r$) (Mettig et al., 2022) shown in Eq. (2):

$$S_r = (S_a^{-1} + \gamma S_t)^T (S_a^{-1} + \gamma S_t), \tag{2}$$

where $S_a$ is a diagonal matrix containing a priori standard deviations, $S_a^{-1}$ is used as the zeroth-order Tikhonov term, $\gamma$ is a scaling factor, and $S_t$ is the first-order derivative matrix (Rodgers, 2002; Rozanov et al., 2011). The forward model Jacobian matrix ($K$) is needed for the linearization of the ill-posed retrieval problem. $S_y$ is the error covariance matrix and is calculated with the fit residuals from the pre-processing step of the retrieval which corrects for effects not accounted for in the radiative transfer model (RTM) such as the rotational Raman scattering, polarisation correction, and secondary calibration (Mettig et al., 2022). For this study we apply a cloud fraction threshold of <0.3 to avoid scenes with significant cloud coverage. Finally, atmospheric pressure and temperature profiles used in the retrieval are taken from ECMWF ERA-5 model simulations (Hersbach et al., 2020). For more detail about the TOPAS retrieval setup see Table 1 of Mettig et al. (2022).

TOPAS results presented in this study are based on TROPOMI Level 1 (L1) version 2.00 radiances from a pre-operational validation dataset and on CrIS Level 2 (L2) CLIMCAPS (Community Long-term Infrared Microwave Coupled Product System) full spectral resolution version 2 radiance data. The specific TOPAS product applied here has retrievals available for 12 weeks in total between July 2018 to October 2019 which overlaps with the intensive TROPOMI validation measurements made by TOLNet. The 12 weeks of data include retrievals from 2 weeks every 3 months allowing for seasonal validation of TOPAS. Quality control is performed on each retrieval pixel before application in TOPAS as described in detail in Mettig et al. (2021, 2022).

### 2.3.1 TROPOMI UV-only retrievals

TROPOMI is a nadir-viewing spectrometer which was launched in October 2017 and has an equatorial overpass time ~13:30 (local time) and a swath width of ~2,600 km providing near daily global coverage. TROPOMI makes retrievals

in the UV (270–330 nm), VIS (320–500 nm), near infrared (NIR, 675–775 nm), and shortwave infrared (SWIR, 2305–2385 nm) (Veefkind et al., 2012). The UV spectrometer has 0.5 nm spectral resolution and 0.065 nm sampling. Vertical profiles of $O_3$ are retrieved using two bands of UV wavelengths [i.e., UV1 (270–300 nm) and UV2 (300–330 nm)] with nadir spatial resolutions of $28.8 \times 5.6$ km$^2$ (cross track $\times$ along track) and $3.6 \times 5.6$ km$^2$, respectively. In order to be combined with the coarser data from CrIS, TROPOMI UV retrievals are degraded to the spatial resolution

of $42 \times 42$ km$^2$. The TROPOMI TOPAS UV-only wavelength retrieval is described in detail in Mettig et al. (2021). The RTM used to simulate TROPOMI retrievals in the UV1 and UV2 wavelengths is SCIATRAN-V4.5 (Rozanov et al., 2011) with the assumption of a pseudo-spherical atmosphere and $O_3$ absorption cross sections from Serdyuchenko et al. (2014). For more detail about the TROPOMI retrieval setup see Table 2 of Mettig et al. (2022).

### 2.3.2 TROPOMI/CrIS UV+TIR retrievals

CrIS retrievals and combined retrievals from TROPOMI and CrIS were produced and described in detail in Mettig et al. (2022). CrIS is a Fourier-transform spectrometer launched in October 2011 and retrieves soundings in the TIR covering the SWIR (3.92–4.64 μm), middle-wave (MWIR, 5.71–8.26 μm), and long-wave infrared (LWIR, 9.14–15.38 μm) (Han et al., 2013; Strow et al., 2013) with a spectral resolution of 0.625 cm$^{-1}$. Ozone retrievals from the University of Bremen algorithm uses a continuous spectrum from 9350 and 9900 nm with a spectral sampling of 0.05

260 nm. The same RTM [SCIATRAN-V4.5 (Rozanov et al., 2011)] is applied to model the radiances in both UV and IR ranges. It is possible to combine observations from TROPOMI and CrIS since Suomi-NPP is in the same orbit as S5P and there is only a 3-minute offset in overpass times. For the $O_3$ profiles, the CrIS field-of-view consisting of $3 \times 3$ circular pixels, each with 14 km diameter at nadir, are combined resulting in a spatial resolution of $42 \times 42$ km$^2$. For more detail about the CrIS retrieval setup see Table 3 of Mettig et al. (2022).

**2.4 Evaluation technique**

TOPAS $O_3$ profile retrievals using TROPOMI UV, CrIS IR, and TROPOMI/CrIS UV+IR data were evaluated using TOLNet and ozonesonde observations. The satellite retrievals were compared to raw observations and when convolved ($X_c$) with the averaging kernel (**AK**) and a priori information from each retrieval using Eq. (3):

$$X_c = X_a + \textbf{AK}(X_t - X_a), \tag{3}$$

where $X_a$ is the a priori $O_3$ profile, $X_t$ is the TOLNet/ozonesonde data interpolated (linear) to the vertical resolution of TOPAS, and **AK** is the averaging kernel matrix. The TOPAS retrieval is conducted with relative deviations from the $X_a$, therefore the **AK** is converted appropriately as explained in Mettig et al. (2021). Statistical comparisons between co-located satellite retrievals and observations were conducted using spatiotemporal thresholds of 2.5 hours and 30 km. Sensitivity studies were conducted using coarser co-location spatiotemporal thresholds of 5 hours and 100

275 km to maximize the number of co-locations for statistical evaluation and to be more consistent with recent TROPOMI/CrIS $O_3$ profile validation studies which use looser colocation thresholds (Mettig et al., 2021, 2022). As

this study focuses on tropospheric $O_3$ which has large spatiotemporal variability, we feel the stricter spatiotemporal thresholds are most appropriate.

To intercompare the performance of UV, IR, and UV+IR TOPAS retrievals in idealized/controlled case studies, TOLNet lidar profiles were convolved using AKs and a priori profiles from each of the three retrieval types and a similar calculation as in Eq. (3) except $X_t$ is replaced with a known TOLNet $O_3$ profile (black lines in Fig. 4). The TOLNet profiles are shown in Fig. 4 which represent typical clean atmospheric conditions and events of planetary boundary layer (PBL) pollution enhancements and stratospheric intrusions. The same a priori profile was used for each case to isolate the impact of the different wavelength retrieval AKs. To produce the three cases, a clean/background TOLNet lidar observation from $RO_3QET$ on September 3, 2019 was applied. To perturb this same $O_3$ profile to represent a PBL enhancement and stratospheric intrusion, we multiplied the clean/background TOLNet lidar profile by a factor of 1.5 at and below 3 km asl and between 8 and 18 km asl, respectively. The a priori profile used in Eq. (3) was from the TOPAS retrieval for the clean/background case on September 3, 2019.

The statistical evaluation of co-located satellite data and lidar/ozonesonde observations focused on bias, normalized mean bias (NMB, see Eq. (S1)) which is normalized to the magnitude of observational data convolved with retrieval AKs, root mean squared error (RMSE, see Eq. (S2)), and simple ordinary least-squares linear regression (slope, y-intercept, coefficient of determination ($R^2$)). The evaluation was conducted for the partial column between 0-12 km to represent the troposphere. This vertical extent was chosen as these are the altitudes typically measured by TOLNet. Furthermore, since TOLNet and ozonesonde data provide the unique opportunity to evaluate satellite $O_3$ profiles in the troposphere at various vertical levels, statistics were calculated for 2 km bins between the surface and 12 km asl.

## 3 Results

### 3.1 Intercomparison of the UV, IR, and UV+IR retrievals

The AK is an important aspect of satellite retrievals and illustrates the sensitivity of the satellite measurement at any altitude to the true atmosphere (Rodgers, 2002). Examples of the mean AKs of all three retrievals are shown in Fig. 2. Each of the three retrievals display different total column DOFs (0-60 km asl) with UV+IR retrievals having the highest sensitivity (5.65) followed by UV-only (5.01) and IR-only (2.28). For all the retrievals the UV-only wavelengths have the largest sensitivity to $O_3$ in the stratosphere above 25 km. Below 20 km, in particular between 10-20 km, IR-only AKs are larger by up to a factor of two compared to UV-only retrievals. While IR retrievals have limited sensitivities above 25 km, it greatly improves the tropospheric sensitivity of UV+IR retrievals compared to UV-only in the troposphere. While total column DOFs in UV+IR retrievals are only slightly larger compared to UV-only, UV+IR retrievals have ~33% larger tropospheric DOFs (0-12 km asl) compared to UV-only as shown in Fig. 2. The increases in total column and tropospheric DOFs can be even larger than this at certain times and locations as demonstrated in Mettig et al. (2022). While all retrievals have minimal sensitivity to the lowermost troposphere (0-2 km asl), UV+IR AKs have ~50% higher DOFs in the lower portion of the atmosphere compared to UV-only retrievals. A similar study applying UV+IR retrievals except from GOME-2+IASI (Cuesta et al., 2013) showed even larger

increases in sensitivity in the troposphere due to the addition of IR wavelengths in comparison to TROPOMI+CrIS used in this study and Mettig et al. (2022). This demonstrates that different combinations of joint UV+IR satellite retrievals can have varying impacts on the sensitivity to $O_3$ in the troposphere.

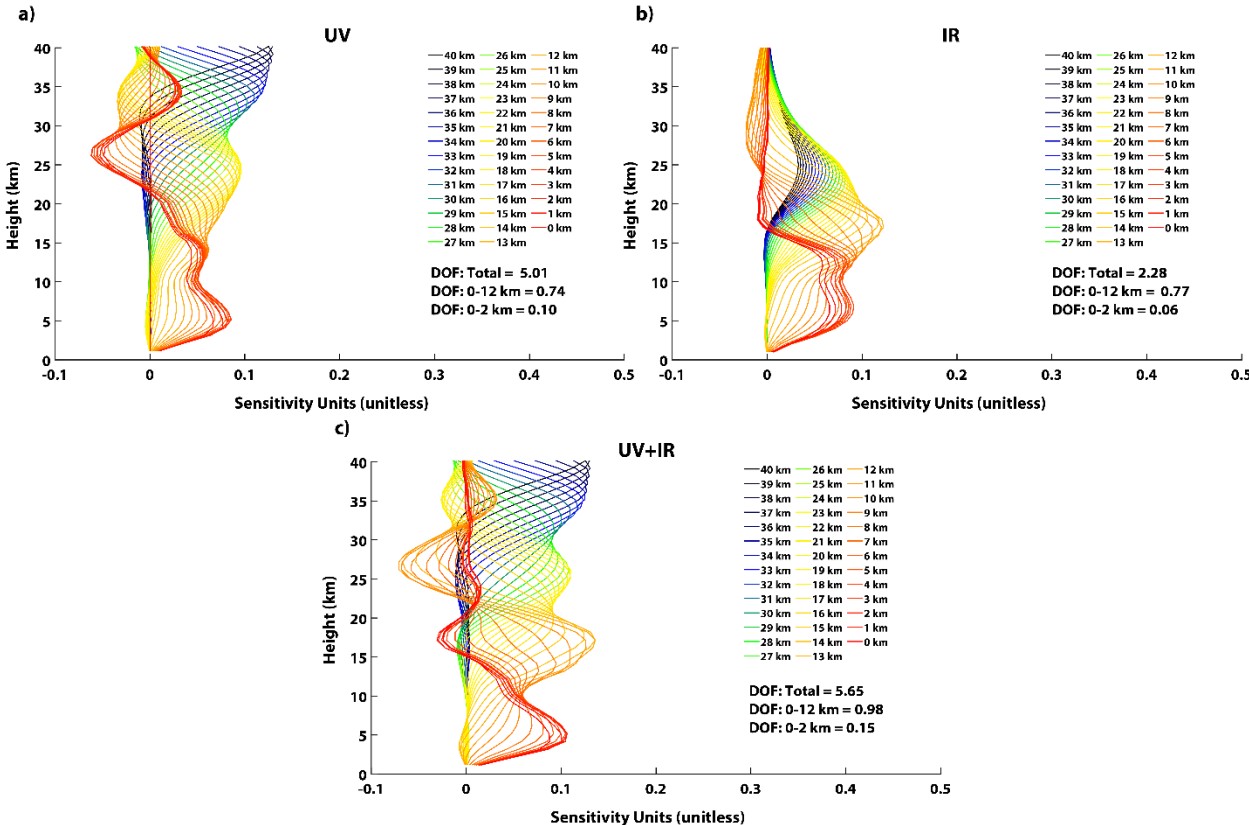

**Figure 2. Mean AKs for all a) UV-only, b) IR-only, and c) UV+IR retrievals applied in this study at the location of RO$_3$QET (34.73 °N, 86.65 °W). Each line shows the AK for a particular 1 km vertical bin from the surface to 40 km asl. The figure inset shows the DOF values for the entire atmosphere (Total, 0-60 km), 0-12 km partial tropospheric column, and 0-2 km lowermost tropospheric partial column.**

The vertical resolution of the retrieval is calculated by the inverse of the diagonal of the AK matrix (Rodgers, 2002) and an example for each retrieval is shown in Fig. 3. UV-only retrievals have the highest vertical resolution (between 10-12 km) above 20 km in the stratosphere. Below this altitude the UV-based retrievals have decreased vertical resolution (~20 km) with coarsest resolution at altitudes around 15 and 10 km asl. This suggests that UV-only $O_3$ profiles have limited information from the retrieval in the mid- to upper-troposphere. IR-only retrievals have limited information above 30 km asl as vertical resolution and AK values are diminished. However, below 20 km asl, IR-only retrievals have higher vertical resolution compared to UV-only data. Between 5-15 km asl IR-only retrievals have vertical resolutions as low as ~12 km. When combining UV and IR information vertical resolutions of the retrievals are improved (8-10 km) compared to IR-only above 12 km and below 8 km.

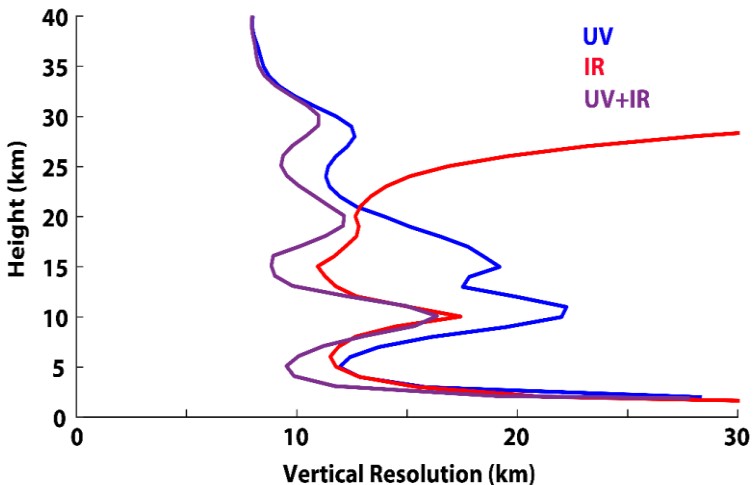

**Figure 3. Average vertical resolution of the UV-only (blue), IR-only (red), and UV+IR (purple) retrievals at the location of RO3QET (34.73 °N, 86.65 °W) for each 1 km vertical bin from the surface to 40 km asl.**

**3.2 Intercomparison of UV, IR, and UV+IR convolved lidar profiles**

To evaluate all three retrievals in idealized and controlled case studies, we produced example TOPAS retrievals by convolving known lidar profiles with each retrieval's AK and a priori profile using Eq. (3) for three scenarios: a)

335 background/clean conditions, b) PBL pollution enhancement, and c) stratospheric intrusion (see description in Sect. 2.4). The results of this intercomparison for the three case studies are shown in Fig. 4. From Fig. 4a it can be seen that despite the a priori profile having a low bias throughout the troposphere compared to the truth in the clean/background conditions case, the true lidar profiles convolved with AKs from all three retrievals are able to accurately reproduce the truth. In the partial column covering the majority of the troposphere (0-12 km asl, hereinafter referred to as the

340 tropospheric column), all example retrieval profiles have minimal biases <3 ppb (absolute value of NMB ≤3%) where the a priori profile had a low bias of ~-16 ppb (NMB = -27%). This suggests that all three retrievals are able to retrieve tropospheric column $O_3$ abundance regardless of biases in the a priori profile for typical background/clean conditions. Similar to the tropospheric column, the 0-2 km partial column (hereinafter referred to as the lowermost tropospheric column) was well reproduced by the true lidar profile convolved by all three retrieval AKs with absolute value NMBs

≤6%.

It is most interesting to see how retrievals perform in physicochemical environments which differ from typical clean/background conditions (e.g., pollution events, stratospheric intrusions, wildfires). Figure 4b and 4c demonstrate whether the UV-only, IR-only, and UV+IR retrievals were able to replicate tropospheric and lowermost tropospheric $O_3$ during a PBL pollution event and a stratospheric intrusion, respectively. During the PBL $O_3$

enhancement event, results of the convolution of the known lidar profiles with AKs from UV-only, IR-only, and UV+IR wavelength retrievals were similar to those for clean/background conditions with only slightly larger values. This small adjustment is due to these retrievals having minimal sensitivity in the lowermost troposphere. For all example retrievals low biases in the lowermost troposphere of ~-40% were seen. Regardless of the inability of the retrievals to fully capture the PBL $O_3$ enhancement, tropospheric column biases for all true lidar profiles representative

of the three retrievals had absolute values ≤13%. Overall, in comparison to the a priori profile, the three example retrievals result in smaller biases throughout the troposphere. This suggests that the retrievals provide some

information for studying large pollution events; however, the limited sensitivity to the lowermost troposphere largely limits these retrievals.

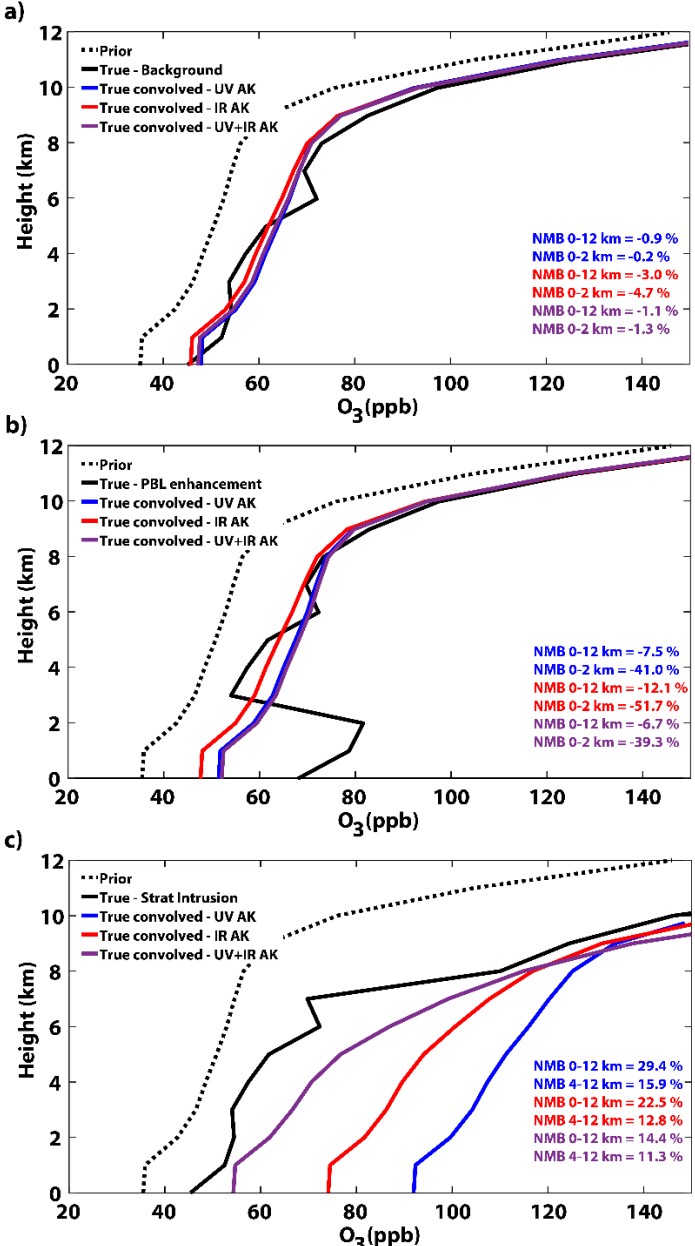

**Figure 4. Example TOPAS retrievals produced from TOLNet lidar profiles convolved with AKs and a priori profiles from UV-only (blue), IR-only (red), and UV+IR (purple) retrievals at the location of RO₃QET (34.73 °N, 86.65 °W) from the surface to 40 km asl for the case studies of: a) clean/background conditions, b) PBL pollution enhancement, and c) stratospheric intrusion. The figure inset shows the normalized mean bias (NMB) percent for the 0-12 tropospheric and 0-2 lowermost tropospheric (Fig. 4b) and mid- to -upper-tropospheric (4-12 km, Fig. 4c) partial column. The reference used to calculate the NMB is the "true" atmospheric state provided by the TOLNet observations.**

For a stratospheric intrusion, the true lidar profiles convolved with retrieval AKs had high biases compared to the truth throughout the troposphere (see Fig. 4c). The large $O_3$ concentrations in the middle to upper troposphere, where all three retrievals have some sensitivity to the true atmosphere, results in NMBs between 14.4% and 47.2%

for tropospheric columns. Regardless of the high biases in the convolved lidar profiles, they still evaluate better for tropospheric column abundances compared to the a priori which had NMB = -51.2%. Compared to the truth, the example UV+IR retrievals replicate these dynamic $O_3$ profiles throughout the troposphere with the most skill. In the mid- to upper-troposphere (4-12 km), UV+IR retrievals had the least high bias (NMB) of 11.3% while IR-only (12.8%) and UV-only (15.9%) retrievals had larger high biases. Compared to the a priori, true lidar profiles convolved with all three retrieval AKs compared much more accurately emphasizing the ability of these retrievals to capture enhanced mid- to -upper tropospheric $O_3$ enhancements. Only the UV+IR example retrieval was able to replicate lowermost tropospheric $O_3$ (NMB = 10.9%) better compared to the a priori (NMB = -33.9%). This sensitivity study suggests that retrievals other than UV+IR data may be challenged to accurately observe $O_3$ profiles throughout the entire tropospheric column during times of enhanced middle- and upper-tropospheric $O_3$ concentrations.

This analysis of complex atmospheric environments important for air quality using idealized retrievals, produced with known $O_3$ profiles convolved separately with different retrieval AKs, in this study expands on past studies that have evaluated TROPOMI/CrIS retrievals (Mettig et al., 2022; Malina et al., 2022). It is important to understand the extent to which TROPOMI, CrIS, and TROPOMI/CrIS joint satellite retrievals, which rely on different wavelengths, can accurately retrieve typical and anomalous structures of $O_3$ in the troposphere.

**3.3 Validation of TOPAS UV-only, IR-only, and UV+IR retrievals**

**3.3.1 Mean vertical $O_3$ profile validation**

TOLNet convolved with each retrieval's AK and a priori (observation operator) was the primary data set used to validate satellite retrievals of $O_3$ vertical profiles. Figure 5 shows the comparison of the three vertical $O_3$ profile satellite retrievals to co-located TOLNet observations at the location of all 6 lidars between 2018-2019 (statistics in Table 3). The validation with TOLNet convolved with TOPAS retrieval AKs (hereinafter TOLNet-AK) demonstrates that the TROPOMI UV-only retrieval meets the defined systematic bias requirement of ±30% (ESA, 2014) throughout the troposphere. The CrIS IR-only retrieval of $O_3$ profiles meets the systematic bias threshold requirement of ±10% defined for this spaceborne sensor (JPSS, 2019) from the surface to ~10 km asl. However, above 10 km asl the CrIS IR-only retrievals exceeded the systematic bias requirement threshold. The combined UV+IR retrievals consistently have NMB values lower than ±10% at all altitudes in the troposphere. All three retrievals generally evaluated more consistently to lidar observations compared to the a priori profiles suggesting that the satellite $O_3$ profiles provide useful information for studying tropospheric composition.

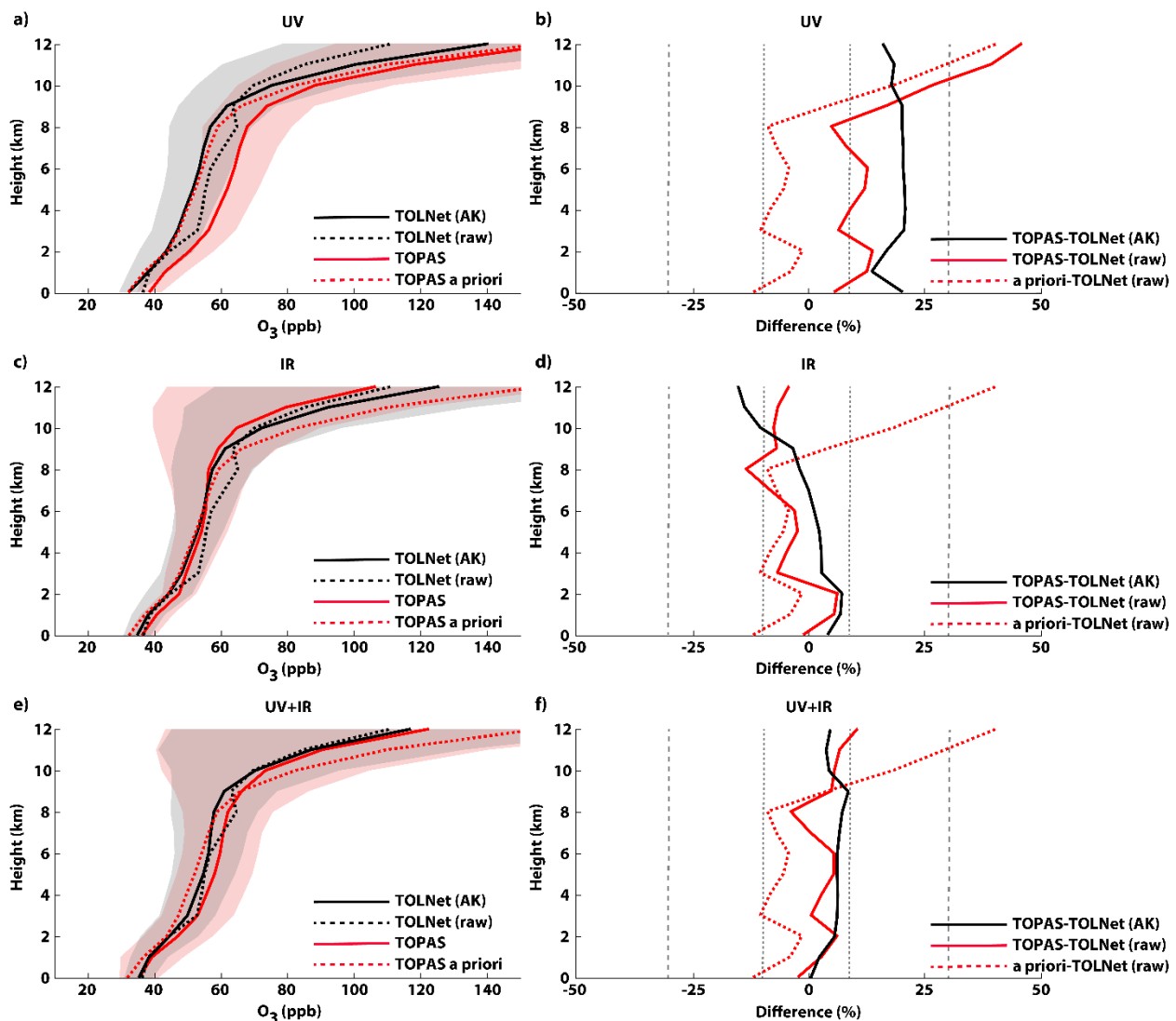

**Figure 5. Vertical O$_3$ profile comparison of TOLNet interpolated to the satellite vertical grid (TOLNet-raw), TOLNet convolved with the TOPAS AK (TOLNet-AK), UV, IR, and UV+IR TOPAS satellite retrievals, and the a priori profile**
**information used in the TOPAS retrieval (total number of colocations (N) = 89). The direct comparison of the profiles and percent difference, displayed as normalized mean bias (NMB), for UV-only (a, c), IR-only (b, d), and UV+IR (e, f) retrievals are displayed, respectively. The NMB between TOPAS satellite retrievals and TOLNet-AK and TOLNet-raw used as the references are labeled as TOPAS-TOLNet (AK) and TOPAS-TOLNet (raw), respectively. The grey and pink shaded regions illustrate the 1σ standard deviation of TOLNet-AK and satellite O$_3$ vertical profiles, respectively. NMB values of 30% and**
**10% are displayed using grey dashed and dotted lines, respectively.**

         The RMSE values in Table 3 represent the random errors in the daily TOPAS O$_3$ profile retrievals when validated with TOLNet-AK observations. While systematic biases were significantly reduced in the three retrievals compared to the a priori profiles, random errors still remained elevated in most instances. UV-only retrievals had unresolved errors ~35% less compared to the a priori. However, the average RMSE values for this retrieval product
still remained large (~17 ppb) throughout the troposphere (compared to ~27 ppb for the a priori). IR-only retrievals displayed the least unresolved errors of all three retrievals with average RMSE values throughout the troposphere of ~10.5 ppb which is ~60% less compared to the a priori. The combined UV+IR profiles had average RMSE values ~14 ppb, ~50% less compared to the a priori, throughout the troposphere. The fact that unresolved errors are reduced in

all three retrievals compared to the a priori further emphasizes that satellite $O_3$ profiles can provide useful information in the troposphere. However, given that unresolved errors of daily profiles on average still remain large (>10 ppb) the accuracy of these satellite products still suffer due to the limited sensitivity of spaceborne sensors to tropospheric $O_3$. These RMSE values calculated with TOLNet-AK are similar in magnitude to those derived using ozonesonde data to validate TROPOMI/CrIS retrievals in Malina et al. (2022) in the troposphere (at 681 mb). Overall, the improvement in tropospheric $O_3$ retrievals from IR-only, and joint UV+IR retrievals, compared to UV-only determined through the validation with TOLNet agrees with many recent studies (e.g., Landgraf and Hasekamp, 2007; Worden et al., 2007b; Cuesta et al., 2013, 2018; Costantino et al., 2017; Colombi et al., 2021; Malina et al., 2022; Mettig et al., 2022).

While observations convolved with the observation operator is the primary validation data source, comparing the three retrievals to TOLNet observations not convolved with the retrieval AKs (hereinafter TOLNet-raw) is also important to understand how the satellite retrievals reproduce actual atmospheric composition in the troposphere. The comparison of TOPAS retrievals and TOLNet-raw suggests that the TROPOMI UV-only data has NMB lower than ±15% at all altitudes below 10 km asl with high biases >30% between 10-12 km asl. A general low bias in IR-only profiles compared to TOLNet-raw is determined below 12 km asl (NMB typically between -5 and -15%) except for ~ 2 km asl where a slight high bias is calculated. The combined UV+IR retrievals compare most closely to TOLNet-raw observations with NMB lower than ±10% at all altitudes. Overall, the IR-only and UV+IR satellite retrieval products evaluate more favorably to TOLNet-AK data compared to TOLNet-raw. However, UV-only profiles have higher biases when evaluated with TOLNet-AK data compared to TOLNet-raw below 10 km asl. To have a more consistent validation of the three $O_3$ profile retrievals we primarily used TOLNet-AK throughout the rest of the study.

In the troposphere, the UV-only retrievals are consistently biased high (NMB = 15-20%) compared to TOLNet-AK lidar data (see Fig. 5a, b; Table 3). The systematic high bias determined in this study agrees with the recent TOPAS validation study by Mettig et al. (2022) and the validation of TROPOMI/CrIS in Malina et al. (2022). Due to Rayleigh scattering, UV-only retrievals are most sensitive to the stratosphere and limited in the troposphere (Chance et al., 1997). IR-only $O_3$ profiles have a small high bias in the lowest 8 km asl and a systematic low bias up to -12% above this altitude. This low bias in the middle to upper troposphere determined in this study agrees with the recent work by Mettig et al. (2022) and to a lesser extent from that found in Malina et al. (2022). Finally, the UV+IR retrievals have minimal bias throughout the troposphere with NMB values ranging from 1% to 8% demonstrating minimal dependance on altitude which also agrees with past TROPOMI/CrIS validation studies (Mettig et al., 2022; Malina et al., 2022). The agreement in the validation statistics of TROPOMI UV, CrIS IR, and TROPOMI/CrIS UV+IR retrievals determined in this study when using TOLNet-AK and those using primarily ozonesonde data (Mettig et al., 2022; Malina et al., 2022) demonstrates that TOLNet is a sufficient validation source for satellite $O_3$ profile retrievals in the troposphere.

**Table 3. Statistical validation of TOPAS UV, IR, and UV+IR retrievals with convolved TOLNet-AK observations. All observations and satellite retrievals were co-located using a 2.5 hour and 30 km threshold criteria.**

**Prior**

| Vertical Level | N (#) | Bias (ppb) | NMB (%) | RMSE (ppb) | Slope |
|---|---|---|---|---|---|
| 0-2 km | 91 | -1.7 | -8.0 | 14.2 | 0.05 |
| 2-4 km | 172 | -5.1 | -6.0 | 14.5 | -0.02 |
| 4-6 km | 172 | -2.8 | -7.0 | 12.5 | 0.09 |
| 6-8 km | 159 | -5.0 | -5.8 | 18.3 | 0.16 |
| 8-10 km | 126 | 7.3 | -2.4 | 29.9 | 0.19 |
| 10-12 km | 84 | 34.9 | 23.8 | 62.8 | 0.82 |
| Trop. Column | 804 | 1.9 | -1.0 | 26.8 | 0.47 |

**UV-only**

| Vertical Level | N (#) | Bias (ppb) | NMB (%) | RMSE (ppb) | Slope |
|---|---|---|---|---|---|
| 0-2 km | 91 | 6.2 | 16.3 | 10.4 | 0.47 |
| 2-4 km | 172 | 9.6 | 18.0 | 14.6 | 0.14 |
| 4-6 km | 172 | 10.4 | 20.1 | 16.0 | 0.20 |
| 6-8 km | 159 | 10.9 | 19.7 | 16.8 | 0.47 |
| 8-10 km | 126 | 12.5 | 19.5 | 18.2 | 0.80 |
| 10-12 km | 84 | 19.6 | 17.5 | 28.2 | 1.02 |
| Trop. Column | 804 | 11.2 | 18.5 | 17.4 | 0.96 |

**IR-only**

| Vertical Level | N (#) | Bias (ppb) | NMB (%) | RMSE (ppb) | Slope |
|---|---|---|---|---|---|
| 0-2 km | 91 | 2.6 | 5.4 | 6.3 | 0.61 |
| 2-4 km | 172 | 1.3 | 4.9 | 6.5 | 0.54 |
| 4-6 km | 172 | 0.8 | 2.4 | 7.5 | 0.62 |
| 6-8 km | 159 | -0.5 | 0.5 | 8.4 | 0.76 |
| 8-10 km | 126 | -4.6 | -2.7 | 12.2 | 0.90 |
| 10-12 km | 84 | -15.9 | -12.1 | 21.2 | 0.89 |
| Trop. Column | 804 | -1.7 | -0.3 | 10.5 | 0.97 |

**UV+IR**

| Vertical Level | N (#) | Bias (ppb) | NMB (%) | RMSE (ppb) | Slope |
|---|---|---|---|---|---|
| 0-2 km | 91 | 1.5 | 1.3 | 10.1 | 0.57 |
| 2-4 km | 172 | 3.2 | 5.8 | 11.7 | 0.46 |
| 4-6 km | 172 | 3.4 | 6.2 | 12.3 | 0.45 |
| 6-8 km | 159 | 4.0 | 6.4 | 11.9 | 0.61 |
| 8-10 km | 126 | 4.2 | 7.8 | 15.6 | 1.00 |
| 10-12 km | 84 | 4.4 | 4.1 | 23.2 | 1.03 |
| Trop. Column | 804 | 3.5 | 5.3 | 14.0 | 0.92 |

Ozonesondes, which are a commonly-used validation data source for evaluating satellite $O_3$ profile retrievals, were also used to validate TOPAS $O_3$ retrievals. Figure 6 shows the comparison of the three vertical $O_3$ profile satellite retrievals to co-located ozonesonde observations at the locations displayed in Table 2 between 2018-2019 (statistics in Table 4). Ozonesondes convolved with retrieval AKs and a priori profiles (hereinafter Ozonesonde-AK) when compared to TROPOMI UV-only retrievals suggest these retrievals meet the defined systematic bias requirement of $\pm30\%$ (ESA, 2014) throughout the troposphere. CrIS IR-only retrievals compared most favorably to ozonesondes meeting the systematic bias requirement of $\pm10\%$ defined for this spaceborne sensor (JPSS, 2019). The combined UV+IR retrievals had NMB values <30% at all altitudes in the troposphere. All three satellite retrieval types performed better than the a priori profile product, further suggesting that the satellite $O_3$ profiles provide useful information for tropospheric composition. Overall, the comparison of TROPOMI UV-only and CrIS IR-only to both TOLNet and ozonesondes suggests these two satellite sensors meet the systematic bias criteria identified for the $O_3$ vertical profile products in the troposphere.

The RMSE values in Table 4 represent the random errors in the daily TOPAS $O_3$ profile retrievals when validated with Ozonesonde-AK observations. All three TOPAS retrievals had lower random errors compared to the a priori profiles; however, random errors still remained elevated in most instances except for the IR-only retrievals. UV-only retrievals had unresolved errors ~50% less compared to the a priori (13.9 ppb). IR-only retrievals displayed the least unresolved errors of all three retrievals with average RMSE values of 6.1 ppb which is ~80% less compared to the a priori. The combined UV+IR profiles had average RMSE values of 11.4 ppb, ~60% less compared to the a priori, throughout the troposphere. Given that unresolved errors of daily profiles on average still remain large (>10 ppb) for retrievals using UV wavelengths (UV, UV+IR), the accuracy of these satellite products still suffer due to the limited sensitivity of spaceborne sensors to tropospheric $O_3$. On the contrary, NMB and RMSE values for IR-only retrievals when compared to Ozonesonde-AK observations were low suggesting this product had some skill in capturing the daily vertical distributions of $O_3$ in the troposphere during this validation. This increased tropospheric sensitivity in IR-only profiles, and when combining UV and IR wavelengths, allows these retrievals to deviate from a biased a priori profiles which improves the ability of this retrieval to capture daily $O_3$ vertical profile distribution variability in the troposphere which is agreement with many recent studies (e.g., Landgraf and Hasekamp, 2007; Worden et al., 2007b; Cuesta et al., 2013, 2018; Costantino et al., 2017; Colombi et al., 2021; Malina et al., 2022; Mettig et al., 2022).

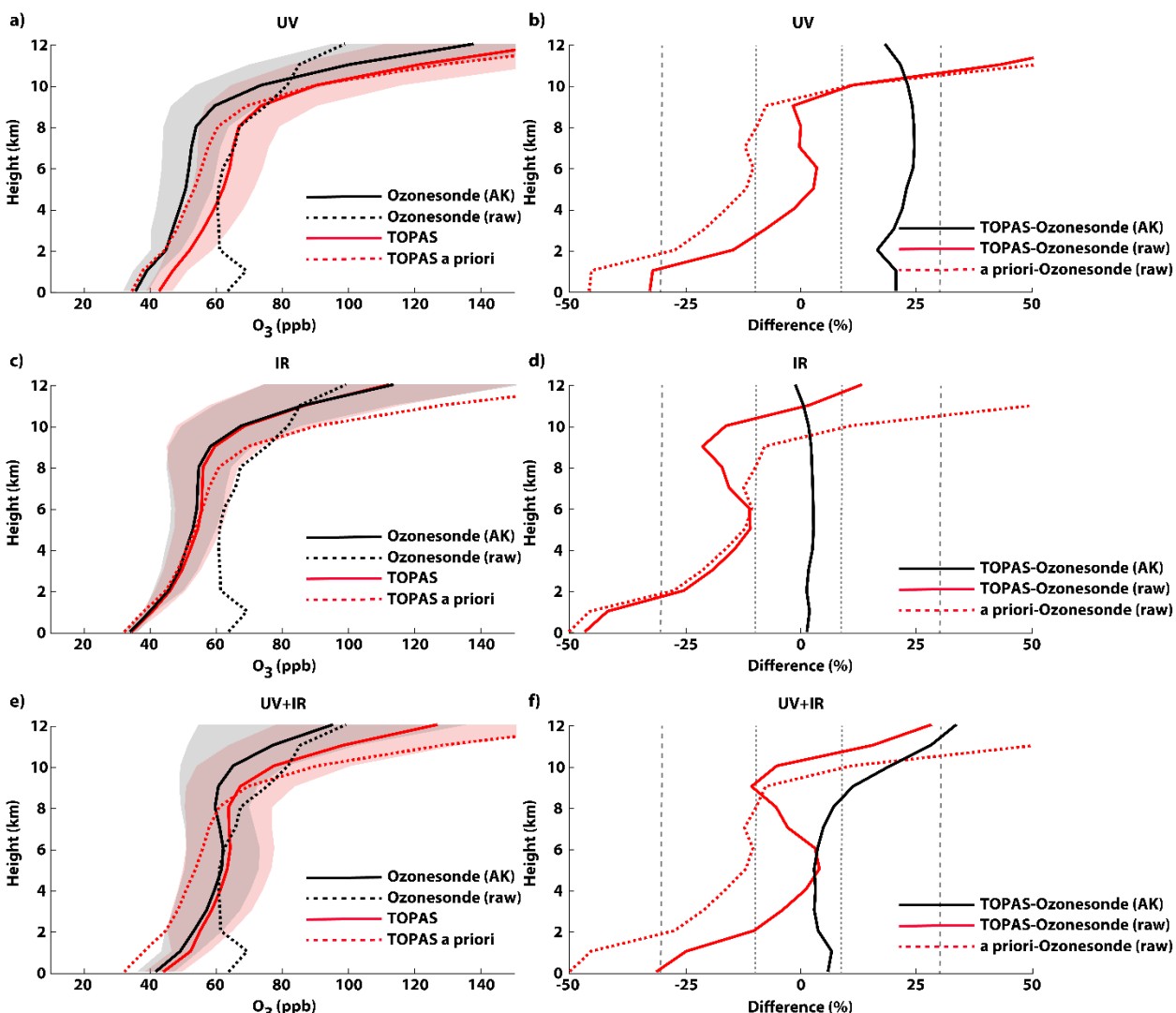

**Figure 6. Vertical O₃ profile comparison of ozonesondes interpolated to the satellite vertical grid (Ozonesonde-raw), ozonesondes convolved with the TOPAS AK (Ozonesonde-AK), UV, IR, and UV+IR TOPAS satellite retrievals, and the a priori profile information used in the TOPAS retrieval (total number of colocations (N) = 26). The direct comparison of the profiles and percent difference for UV-only (a, c), IR-only (b, d), and UV+IR (e, f) retrievals are displayed, respectively. The normalized mean bias (NMB) between TOPAS satellite retrievals and Ozonesonde-AK and Ozonesonde-raw used as the reference are labeled as TOPAS-Ozonesonde (AK) and TOPAS-Ozonesonde (raw), respectively. The grey and pink shaded regions illustrate the 1σ standard deviation of Ozonesonde-AK and satellite O₃ vertical profiles, respectively. NMB values of 30% and 10% are displayed using grey dashed and dotted lines, respectively.**

Comparing the three satellite profile retrievals to ozonesonde observations not convolved with the retrieval AKs (hereinafter Ozonesonde-raw) suggests TROPOMI UV-only data has NMB lower than ±10% between 3 and 10 km asl; however, much higher biases above and below these altitudes (see Fig. 6). A low bias in IR-only profiles compared to Ozonesonde-raw was determined below 11 km asl (NMB between -10 and -50%) with a small positive bias above this altitude. The combined UV+IR retrievals compare most closely to Ozonesonde-raw observations with NMB lower than ±10% at all altitudes below 11 km and above 2 km asl. The low bias in all three satellite retrievals in the lowermost troposphere is caused by the lack of sensitivity to O₃ at these altitudes not allowing the retrievals to replicate the larger O₃ concentrations observed in the PBL by Ozonesonde-raw data. The evaluation of TOPAS

retrievals with Ozonesonde-raw differs from results using TOLNet-raw primarily below 4 km where ozonesonde observed large $O_3$ enhancements in the lowermost troposphere which were not evident in the TOLNet data. In general, the satellite retrievals compare more consistently to Ozonesonde-AK instead of Ozonesonde-raw.

In the troposphere, UV-only retrievals were consistently biased high compared to Ozonesonde-AK data (see Fig. 6a, b; Table 4). This systematic high bias is consistent with the validation using TOLNet-AK observations. IR-

only $O_3$ profiles compare very well to Ozonesonde-AK data with NMB values <3% throughout the troposphere. This outperforms the IR-only profiles when compared to TOLNet-AK data which displayed a low bias aloft. Finally, the UV+IR retrievals have minimal bias below 10 km asl with NMB values <10% when compared with TOLNet-AK observations; however, when compared with Ozonesonde-AK data the UV+IR retrievals had a noticeable high bias above 9 km. The overall validation of the three satellite $O_3$ profile retrievals using Ozonesonde-AK was generally

consistent compared to when using TOLNet-AK. It is important to note that TOLNet and ozonesonde validation statistics are generally consistent given the fact that ozonesondes are a highly-accurate and commonly-applied satellite validation data source. This suggests that TOLNet is a sufficient validation data source of tropospheric $O_3$ profile retrievals from satellites. Given that TOLNet is able to accurately validate satellite-derived $O_3$ profiles, and the focus of this work is on the demonstration of TOLNet for validating satellite retrievals, the rest of this study focuses on the

validation using the lidar network observations.

**Table 4. Statistical validation of TOPAS UV, IR, and UV+IR retrievals with convolved ozonesonde observations. All observations and satellite retrievals were co-located using a 2.5 hour and 30 km threshold criteria.**

**Prior**

| Vertical Level | N (#) | Bias (ppb) | NMB (%) | RMSE (ppb) | Slope |
|---|---|---|---|---|---|
| 0-2 km | 49 | -21.8 | -45.7 | 28.2 | -0.01 |
| 2-4 km | 47 | -10.4 | -24.2 | 12.5 | 0.26 |
| 4-6 km | 50 | -8.3 | -14.2 | 12.8 | 0.25 |
| 6-8 km | 50 | -5.8 | -11.4 | 13.9 | 0.33 |
| 8-10 km | 50 | 4.7 | -8.7 | 18.8 | 0.71 |
| 10-12 km | 50 | 64.4 | 30.8 | 75.3 | 1.28 |
| Trop. Column | 296 | 1.5 | -12.2 | 29.5 | 0.78 |

**UV-only**

| Vertical Level | N (#) | Bias (ppb) | NMB (%) | RMSE (ppb) | Slope |
|---|---|---|---|---|---|
| 0-2 km | 49 | 7.4 | 20.2 | 8.6 | 0.88 |
| 2-4 km | 47 | 9.8 | 18.0 | 11.5 | 0.90 |
| 4-6 km | 50 | 11.8 | 22.1 | 14.1 | 0.75 |
| 6-8 km | 50 | 12.8 | 24.1 | 15.1 | 0.87 |
| 8-10 km | 50 | 15.4 | 24.0 | 19.0 | 1.12 |
| 10-12 km | 50 | 22.9 | 22.1 | 32.1 | 1.07 |
| Trop. Column | 296 | 12.9 | 21.8 | 13.9 | 1.01 |

**IR-only**

| Vertical Level | N (#) | Bias (ppb) | NMB (%) | RMSE (ppb) | Slope |
|---|---|---|---|---|---|
| 0-2 km | 49 | 0.5 | 1.6 | 3.8 | 0.81 |
| 2-4 km | 47 | 1.1 | 1.5 | 4.8 | 0.71 |
| 4-6 km | 50 | 1.5 | 2.7 | 5.7 | 0.74 |
| 6-8 km | 50 | 1.4 | 2.7 | 6.8 | 0.80 |
| 8-10 km | 50 | 1.2 | 2.4 | 12.0 | 0.86 |
| 10-12 km | 50 | -0.3 | 1.2 | 29.3 | 0.66 |
| Trop. Column | 296 | 0.8 | 2.0 | 6.1 | 0.85 |

**UV+IR**

| Vertical Level | N (#) | Bias (ppb) | NMB (%) | RMSE (ppb) | Slope |
|---|---|---|---|---|---|
| 0-2 km | 49 | 2.3 | 6.1 | 6.7 | 0.79 |
| 2-4 km | 47 | 1.7 | 3.2 | 8.0 | 0.91 |
| 4-6 km | 50 | 1.9 | 2.9 | 8.1 | 0.96 |
| 6-8 km | 50 | 3.5 | 4.1 | 8.2 | 1.03 |
| 8-10 km | 50 | 9.5 | 9.0 | 17.0 | 1.05 |
| 10-12 km | 50 | 26.4 | 23.5 | 47.8 | 0.64 |
| Trop. Column | 296 | 7.2 | 8.1 | 11.4 | 0.93 |

**3.3.2 Impact of co-location criteria on mean vertical O$_3$ profile validation**

To determine the impact of using coarser spatiotemporal co-location criteria (5 hours and 100 km) for satellite O$_3$ profile validation, more consistent with recent TROPOMI/CrIS validation studies (Malina et al., 2022; Mettig et al., 2022), we conducted a sensitivity study validation of the mean vertical O$_3$ profiles using the coarser co-location criteria. Figure S1 shows the comparison of the three vertical O$_3$ profile satellite retrievals to co-located TOLNet-AK observations at the location of all 6 lidars between 2018-2019 using the coarser spatiotemporal co-location criteria (statistics in Table S1). The coarser co-location criteria resulted in a larger amount of co-location data points for evaluation. While this results in a more reliable statistical evaluation, the validation of all three satellite retrievals is consistent using both the fine and coarse spatiotemporal co-location criteria. Given the consistent validation, and the fact that representation error between ground-based and satellite data is minimized when applying the finer co-location criteria, including the fact that tropospheric O$_3$ can experience rapid changes during the daylight hours, we feel the finer co-location criteria are more appropriate for this validation.

**3.3.3 Validation at different vertical levels in the troposphere using TOLNet**

A major advantage of using TOLNet for validation of satellite O$_3$ profile retrievals is the ability to make accurate, high temporal and vertical resolution observations at different vertical levels of the troposphere. While TROPOMI and CrIS are polar-orbiting systems which only retrieve O$_3$ profiles once per day, the observations throughout an entire day are vital for validating geostationary sensors such as TEMPO. However, the high vertical resolution and accurate O$_3$ observations from TOLNet are applied here to robustly validate satellite retrievals at multiple layers of the troposphere. Figure 7 shows the direct comparison of all co-located satellite and TOLNet-AK O$_3$ profiles for six separate 2-km vertical layers between the surface and 12 km asl. This TROPOMI/CrIS validation at multiple layers in the troposphere allows for more detailed interpretation of the capability of these satellite vertical profiles to retrieve middle- to lower-tropospheric O$_3$ in comparison to other recent TROPOMI/CrIS validation studies (Malina et al., 2022; Mettig et al., 2022). In the lowest altitudes between the surface and 2 km asl all three retrievals perform similarly; however, the UV-only retrieval has a higher bias and slightly more spread (see bias and RMSE statistics in Table 3). All three retrievals have limited sensitivity to these lower tropospheric regions and are primarily driven by retrieval performance above these altitudes and the shape of the a priori profile. Adding the IR wavelengths, which had NMB of 5.4%, RMSE of 6.3 ppb, and linear regression slope of 0.61, adds additional sensitivity to the lower- to mid-tropospheric regions, improves performance in the UV+IR (NMB of 1.3%, RMSE of 10.1 ppb, slope of 0.57) retrieval compared to the UV-only (NMB of 16.3%, RMSE of 10.4 ppb, slope of 0.47).

Above the lowest portions of the troposphere, the three retrievals have more sensitivity to O$_3$ and differ more in their performance. Between 2-4 km the UV+IR (NMB of 5.8%, RMSE of 11.7 ppb, slope of 0.46) and especially IR-only (NMB of 4.9%, RMSE of 6.5 ppb, slope of 0.54) retrievals outperform UV-only retrievals (NMB of 18.0%, RMSE of 14.6 ppb, slope of 0.14) due to the enhanced sensitivity provided by the IR wavelengths. The UV+IR and IR-only retrievals have better linear regression slopes compared to the UV-only product (UV-only results have similar slopes as the a priori profile below 6 km) due to the ability to deviate further from the a priori profile shape. In the vertical layer between 4-6 km, similar to the layer between 2-4 km, the UV+IR (NMB of 6.2%, RMSE of 12.3 ppb,

slope of 0.45) and in particular the IR-only (NMB of 2.4%, RMSE of 7.5 ppb, slope of 0.62) retrievals outperform UV-only (NMB of 20.1%, RMSE of 16.0 ppb, slope of 0.20) retrievals with less bias and RMSE and better linear regression slopes. It should be noted that the retrievals without UV wavelengths (IR-only) was the only satellite

product with improved statistics (lower NMB and higher slope) at 4-6 km compared to 2-4 km. The vertical level around 4-6 km is where IR-wavelengths have peak sensitivity to $O_3$ in the TOPAS CrIS retrieval, which contributes to this result. However, given that DOFS for $O_3$ profile retrievals are < 1.0 below 12 km agl, no individual 2 km layer evaluated in this study is completely independent from the retrieval performance throughout the troposphere. For the vertical layer between 6-8 km with IR-only retrievals having the best statistical evaluation (NMB of 0.5%, RMSE of

8.4 ppb, slope of 0.76). Overall, between 2-8 km asl, IR-only retrievals have the least bias and spread, along with best linear regression fits. UV+IR retrievals are similar to IR-only data with only slightly worse performance when compared to TOLNet-AK. This result demonstrates that while the combination of UV and IR wavelengths tends to improve the performance of TOPAS retrievals compared to UV-only, this is not always the case for IR-only.

   In the upper troposphere (8-12 km asl), UV-only retrievals still display the largest positive bias (NMB of

17.5–19.5%, RMSE of 18.2–28.2 ppb, slope of 0.80–1.0) compared to UV+IR (NMB of 4.1–7.8%, RMSE of 15.6–23.2 ppb, slope of ~1.0) and IR-only (NMB of -2.7–-12.1%, RMSE of 12.2–21.2 ppb, slope of ~0.9) retrievals. Comparing the spread in the data between all three profile products, the statistical evaluation were more similar in the upper troposphere compared to the altitudes below 8 km asl. All three products have linear regression slopes near unity; however, IR-only retrievals have a noticeable low bias above 8 km asl. In general, the IR-only and UV+IR

retrievals had the least bias out of all three retrievals between 8-10 km and 10-12 km, respectively, when compared to TOLNet-AK and IR-only data has the least spread in the 8-12 km vertical level when compared to observations.

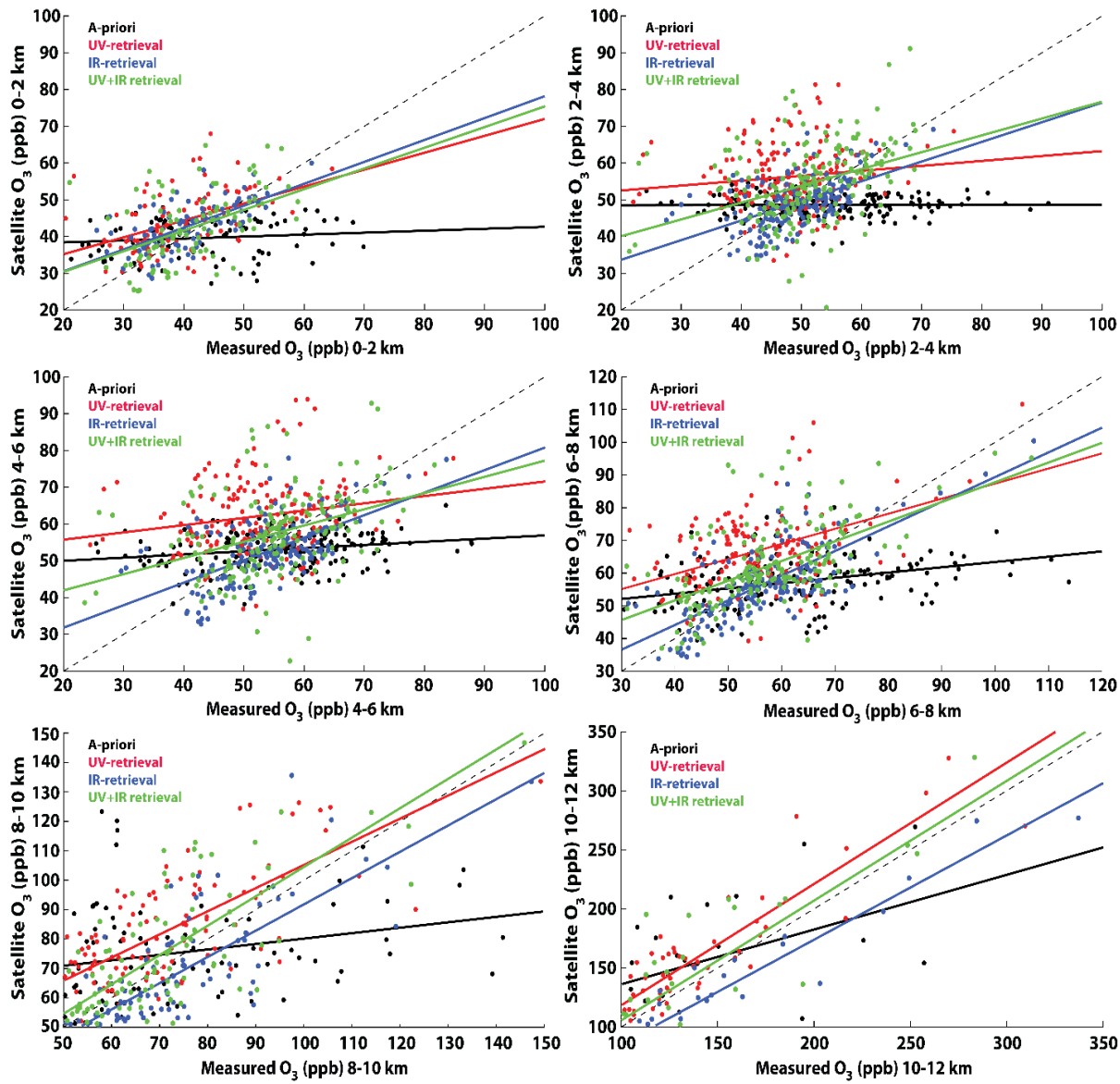

**Figure 7. Scatter plot comparison of co-located TOPAS UV-only (red), IR-only (blue), combined UV+IR (green) retrievals, and a priori $O_3$ vertical profiles to TOLNet observations in 2-km vertical layers between the surface and 12 km asl. The satellite profiles are compared to TOLNet-AK and the a priori data is compared to TOLNet-raw. The solid-colored lines illustrate the linear regression fit of each satellite-TOLNet comparison and the dashed line represents the 1:1 fit line. Statistics of the intercomparison at each vertical level are presented in Table 3.**

At all altitudes in the troposphere the retrievals typically evaluate better (lower bias and RMSE values) to observations compared to the a priori product. The linear regression slope provides information about the capability of the retrieval to deviate from the prior profile shape and magnitudes. Below 8 km the UV-only retrieval has similar linear regression fits compared to the a priori emphasizing the limited sensitivity of these wavelengths to $O_3$ in the lower troposphere. In the lower- to mid-troposphere the IR wavelengths provide additional DOFs which allow the IR-only and UV+IR retrieval to deviate further from the prior profile shape and compare better to observations. Above 8 km all three retrievals have similar linear regression slopes indicating they are able to deviate to some degree from the a priori shape and compare better to observations. While neither of the three retrievals have more than 1.0 DOFs below

12 km asl, the information provided by all retrievals improves upon the prior vertical profile suggesting these satellite data provide useful information for studying tropospheric $O_3$.

### 3.3.4 TOLNet validation of seasonal vertical $O_3$ profiles

A seasonal validation of the three TOPAS $O_3$ profiles retrievals was performed using TOLNet-AK observations. Figure 8 shows the comparison of satellite retrievals and lidar profiles divided into meteorological season (i.e., winter (DJF), spring (MAM), summer (JJA), and fall (SON)). The number of co-locations are limited during the winter and spring (N < 10) outside of the primary $O_3$ season covering the summer and fall. More robust observational coverage by TOLNet is apparent in the summer (N = 34) and fall (N = 41). Given the limited number of co-located observations in the winter and spring available for this study, the statistical validation of the satellite retrievals during these months should be viewed as relatively uncertain. The focus of this study was to demonstrate the capability of TOLNet data to validate satellite retrievals; however, to improve the number of seasonal co-locations we also used ozonesonde data (Ozonesonde-AK), in addition to lidar measurements, and these results are shown in Fig. S2. Given the performance of the validation was similar when using TOLNet-AK and the combination of TOLNet-AK and Ozonesonde-AK, the main text of the paper focuses on the seasonal validation of TOPAS retrievals using TOLNet-AK data only.

During the winter months, UV+IR retrievals compared most closely to TOLNet-AK observations. This combined retrieval is the only satellite product which validates better to observations compared to the a priori below 8 km asl. The NMB of the UV- and IR-only profiles exceed ±10% throughout the majority of the tropospheric column while UV+IR retrievals have NMB values <7% from the surface to 12 km asl. The prior profile and UV-only retrievals have similar unresolved errors/uncertainties (RMSE) of ~24 ppb throughout the tropospheric column, suggesting the UV-only product was unable to improve upon the a priori information. The random errors in the retrievals including IR wavelengths (e.g., IR-only and UV+IR) had lower RMSE values of ~13 ppb.

In the spring months all three retrievals evaluated more consistently to observations compared to the a priori profiles. At all altitudes in the troposphere the IR-only profiles compared the best to observations with NMB values <10%. The two retrievals which incorporate UV wavelengths had larger positive biases compared to the IR-only data with NMB values between 10-15% and 15-25% for the UV+IR and UV-only vertical profiles, respectively. IR-only retrievals in the spring had the lowest bias and random error (RMSE = 7 ppb). UV-only retrievals also had lower random errors compared to the a priori data source (RMSE = 21 ppb) with unresolved errors of ~14 ppb. UV+IR retrievals in the spring had moderate systematic biases and the largest unresolved errors of all three retrievals (~19 ppb).

During the summer, UV+IR had the lowest biases (within ±10%) above 2 km asl compared to other retrievals and the a priori. The UV-only retrievals had a constant systematic high bias of 10-15% throughout the entire troposphere. The IR-only retrievals had variable biases below 8 km while above this altitude displayed a large negative bias. All three retrievals had smaller RMSE values compared to the a priori of 17 ppb, 11 ppb, and 14 ppb for the UV-only, IR-only, and UV+IR retrievals, respectively. Overall, all three satellite retrievals had smaller bias and uncertainties compared to the a priori profiles for the summer months.

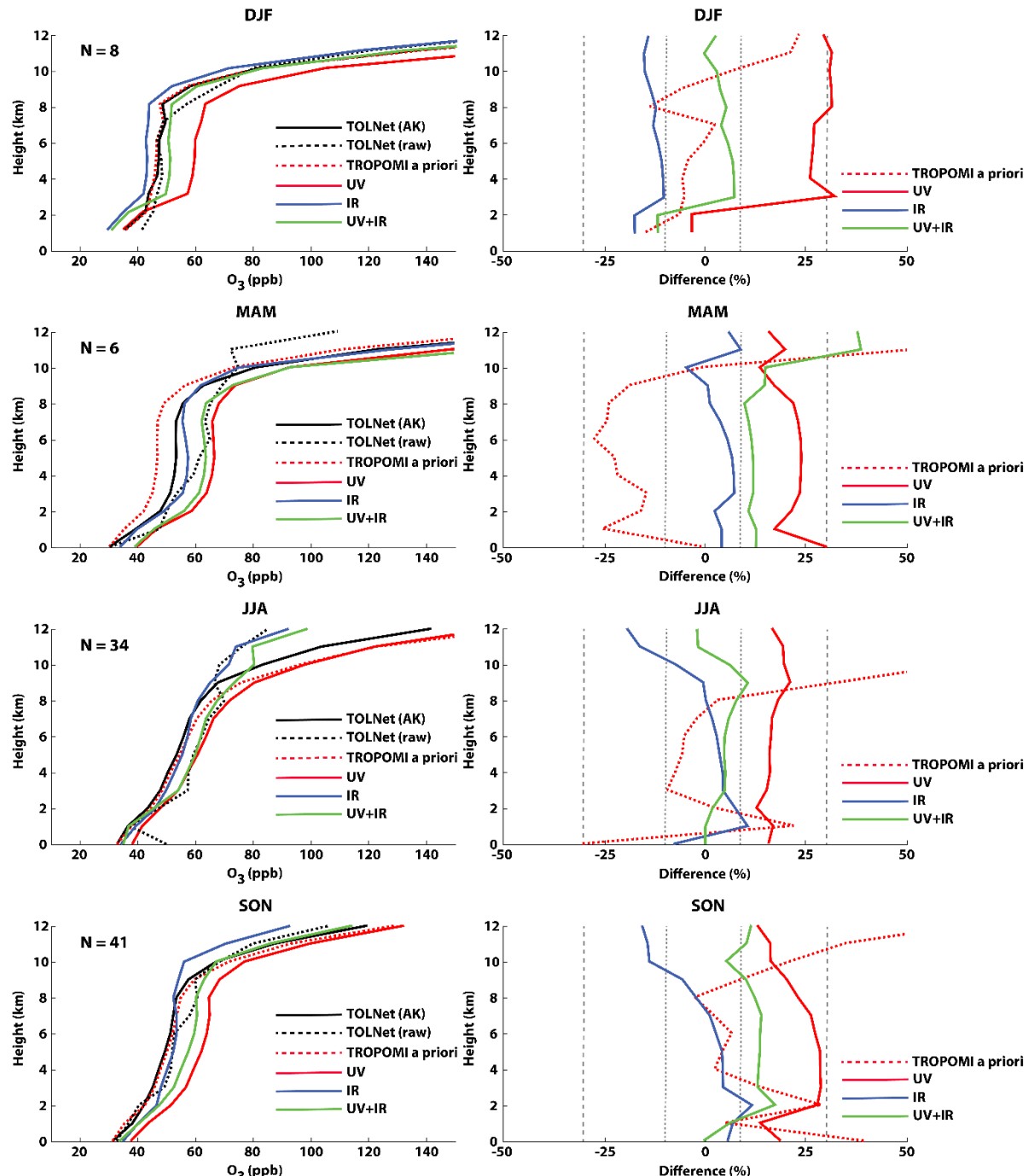

**Figure 8. Seasonally-averaged vertical O₃ profile comparison of TOLNet interpolated to the satellite vertical grid (TOLNet-raw), TOLNet convolved with the TOPAS AKs (TOLNet-AK), UV, IR, and UV+IR TOPAS satellite retrievals, and the a priori profile information. The TOLNet profile convolved with the UV+IR AKs are displayed and the two other (UV- and IR-only) convolved profiles are not shown to reduce the number of profiles presented. The direct comparison of the profiles (left column) and normalized mean bias (NMB) (right column) for UV-only, IR-only, and UV+IR retrievals compared to TOLNet-AK as the reference are displayed, respectively. NMB values for each of the three retrievals are calculated using the TOLNet profiles convolved with the correct retrieval-specific AK as the reference. NMB values of 30% and 10% are displayed using grey dashed and dotted lines, respectively. The total number (N) of co-located profiles are shown in the figure inset.**

625

630

At all altitudes in the troposphere during the fall months the retrievals using IR wavelengths (IR-only and UV+IR) compared the best to observations with NMB values <15%. UV-only retrievals had consistent high biases typically >20%. IR-only profiles had the best overall performance with small biases (within ±10%) below 9 km and a larger negative bias aloft. All three retrievals had smaller RMSE values compared to the a priori of 16 ppb, 10 ppb, and 13 ppb for the UV-only, IR-only, and UV+IR retrievals, respectively. Similar to the summer months, during the fall all three retrievals had noticeably lower random errors compared to the a priori profiles.

The seasonal validation of TROPOMI, CrIS, and TROPOMI/CrIS retrievals in Mettig et al. (2022) and Malina et al. (2022) both did not apply ground-based $O_3$ lidars; therefore, this study provides a new perspective on the seasonal validation of these retrievals. All three TROPOMI, CrIS, and TROPOMI/CrIS validation studies applied a similar number of profiles during each season. Overall, the seasonal validation in Malina et al. (2022) tends to show lower systematic biases and random error compared to this study and Mettig et al. (2022) in all seasons. The satellite retrievals in Malina et al. (2022) were produced using the MUlti-SpEctra, MUlti-SpEcies, MUlti-SEnsors (MUSES) retrieval algorithm which differs in many ways to those produced with the TOPAS algorithm. In general, the validation in this study resulted in lower systematic biases, and bias variability in the mean vertical profiles, in all seasons except for winter months compared to Mettig et al. (2022). All three studies provide useful information about the evaluation of TROPOMI, CrIS, and TROPOMI/CrIS retrievals since they all use different retrieval algorithms to produce satellite data and ground-based observations as validation data sets.

**4 Discussion of retrieval bias and uncertainties**

To determine a retrieval product's accuracy, it is important to quantify systematic bias and random errors. Figure 5 and Table 3 illustrate each of the three retrieval's systematic biases represented by NMB values when validated with TOLNet-AK observations. TOPAS UV-only retrievals have a consistent positive bias ranging between 16-20% throughout the entire troposphere in agreement with the validation of TROPOMI in the middle northern latitudes for the mid- to lower-troposphere by Mettig et al. (2021). These UV-only retrievals had the largest random errors (RMSE) in the troposphere (17.4 ppb) in comparison to IR-only (10.5 ppb) and UV+IR (14.0 ppb) products. The higher random errors derived in this study for UV-only agrees with Mettig et al. (2022) but does not agree with Malina et al. (2022) which derived the lowest values for TROPOMI in comparison to CrIS and TROPOMI/CrIS. In agreement with other recent TROPOMI, CrIS, and TROPOMI/CrIS retrieval validation studies (Mettig et al., 2022; Malina et al., 2022), the addition of the IR-wavelength retrievals to UV-only data was shown to improve the satellite retrievals of $O_3$ profiles throughout the troposphere. Systematic biases from the IR-only retrievals were minimal (NMB <6%) in the lowest 10 km of the troposphere with the product having a negative bias between 10-12 km (NMB = -12%). The low mean biases derived in this study from IR-only retrievals agrees with Mettig et al. (2022) and Malina et al. (2022); however, the negative bias between 10-12 km is unique to this study using lidar profiles for evaluation. The lowest random errors were calculated for CrIS IR-only profiles which agrees with Mettig et al. (2022). When combining the UV and IR wavelengths, the TOPAS retrieval has minimal systematic bias throughout the troposphere with NMB values ranging between 1% and 8% and average RMSE values of 14.0 ppb when validated with TOLNet-AK

observations. These systematic and random errors for UV+IR wavelengths agree well with the work by Malina et al. (2022).

Applying different combinations of UV+IR joint wavelength retrievals (e.g., GOME-2+IASI) also displays improvements compared to UV-only products in the troposphere similar to that determined in this study (e.g., Cuesta et al., 2013, 2018). Cuesta et al. (2013, 2018) demonstrated how GOME-2+IASI retrievals show high accuracy compared to ozonesondes in the lowermost troposphere and displays a clear capability to capture PBL $O_3$ enhancements. This differs from the results of this study which suggest that TROPOMI+CrIS UV+IR joint wavelength retrievals still struggle to reproduce large PBL $O_3$ enhancements due to limited lowermost tropospheric sensitivity. The reasons why GOME-2+IASI displays the remarkable capability to retrieve lowermost tropospheric enhancements compared to the results from TROPOMI+CrIS is not immediately apparent. There are differences in the retrieval algorithms, a priori input data sets, and the spectral resolutions of the UV and IR sensors applied. Comparing our results to Cuesta et al. (2013) shows that DOFs are higher in the troposphere and in the 0-2 km agl column (>33% higher) in GOME-2+IASI retrievals compared to TROPOMI+CrIS which would explain some of the differences in capabilities to retrieve lowermost tropospheric $O_3$ enhancements.

Systematic biases in $O_3$ profile retrievals when compared to raw observations can largely be explained by biases and shape of the a priori vertical profiles (e.g., Kulawik et al., 2006; Zhang et al., 2010; Johnson et al., 2018; Malina et al., 2022). However, this study focuses on systematic and random biases in $O_3$ profile retrievals when compared to observations convolved with individual retrieval's operational operators to remove the impact of a priori profile information. Additional sources of error from sza, surface albedo, and cloud fraction were determined in this study to be controlling factors for systematic bias. All three retrievals had similar bias impacts from sza, surface albedo, and cloud fraction, so here we discuss the analysis of UV+IR retrievals only. When comparing TOPAS UV+IR retrievals to all co-located convolved TOLNet retrievals it was determined that the daily averaged bias was 14.4 ppb. When separating this for high (>60°) and low (<60°) sza it was found that systematic biases were larger (18.3 ppb) for high sza conditions compared to low sza (13.1 ppb). Minimal impact was calculated for RMSE values for high and low sza pixels (~15.0 ppb). The dependance of satellite $O_3$ profile retrievals in the troposphere on sza has also been shown in recent TROPOMI and TOPAS validation studies (e.g., Mettig et al., 2021, 2022). As sza values become large the sensitivity of the retrieval in the troposphere are reduced, leading to increased biases in the satellite products. Surface albedo has also been demonstrated to be a controlling factor for the accuracy of tropospheric $O_3$ retrievals. This validation study using TOLNet observations further confirms this. When separating the TOPAS validation for high (>0.2) and low (<0.2) albedo values it was found that systematic biases were larger (16.4 ppb) for low albedo conditions compared to high surface reflectivity (12.8 ppb). RMSE values, representative of unresolved errors in the retrievals, were larger for low albedo conditions (16.9 ppb) compared to high reflectivity scenes (12.2 ppb). Cloud interference can impact retrievals of most atmospheric constituents such as $O_3$ profiles. Here it was determined that while systematic biases for low cloud scenes (cloud fraction < 0.2) and times of high clouds (cloud fraction > 0.2) were similar (~14 ppb), RMSE values were larger for cloudy scenes (17.1 ppb) compared to clear pixels (13.5 ppb). This study further emphasizes the impact that clouds can have a detrimental impact on the accuracy and uncertainties of $O_3$ profile retrievals.

**5 Conclusions**

This study applied the full complement of TOLNet observations (6 out of 8 systems were operational between 2018-2019) to validate UV-only TROPOMI, IR-only CrIS, and UV+TIR TROPOMI/CrIS TOPAS $O_3$ profile retrievals. TOLNet proved to be a vital validation tool for satellite tropospheric $O_3$ retrievals. TOLNet data provides: a) highly accurate, high temporal resolution, $O_3$ observations for multiple continuous hours and/or days, b) retrievals with minimal dependance on a priori information, and c) profiles with higher vertical resolution compared to satellite products. The multi-hour observations provided by TOLNet will be important for validation of tropospheric $O_3$ profiles and the lowermost tropospheric (0-2 km) partial columns from the recently-launched geostationary TEMPO mission. As a primary validation data source for TEMPO, TOLNet will make dedicated validation observations for this geostationary sensor during all times of the day. These observations will provide hourly tropospheric $O_3$ observations during all seasons which will greatly increase the amount of data from TOLNet needed for seasonal validation which was not available for this study.

TOLNet was used to intercompare the three retrievals, using idealized case studies by convolving high resolution lidar profiles with retrieval-specific AKs of TROPOMI UV, CrIS IR, and TROPOMI/CrIS UV+IR based on the TOPAS algorithm of the University of Bremen. All three retrievals were determined to be able to reproduce typical/background $O_3$ profiles. However, the results differed more for physicochemical environments which deviate from typical clean/background conditions. Retrievals using combinations of wavelengths proved to be more capable of capturing conditions with air quality impacts such as stratospheric intrusions. UV+IR $O_3$ profiles most accurately observed $O_3$ profiles throughout the troposphere during times of enhanced middle- and upper-tropospheric $O_3$ concentrations such as what occurs during stratospheric intrusions. For near-surface $O_3$ pollution conditions, all three retrievals were not able to accurately replicate enhancements in the lowermost troposphere due to minimal sensitivity to this portion of the atmosphere. The reason that combined wavelength retrievals (UV+IR) outperform the single wavelength data products (UV, IR) is the increased vertical resolution and sensitivity to $O_3$ in the troposphere aiding in the ability to deviate further from the a priori profile shape. This analysis of retrieval performance in air quality relevant environments is unique to this study in comparison to other TROPOMI, CrIS, and TROPOMI/CrIS intercomparison studies (e.g., Mettig et al., 2022; Malina et al., 2022).

TOPAS $O_3$ profiles from TROPOMI UV, CrIS IR, and TROPOMI/CrIS UV+IR retrievals were validated with TOLNet and ozonesonde observations. The validation results using the two observational data sets were overall consistent. Compared to TOLNet-AK, UV-only TROPOMI retrievals had mean biases which meet the defined systematic bias requirement of ±30% (ESA, 2014) throughout the troposphere. The CrIS IR-only retrieval of $O_3$ profiles meet the systematic bias requirement of ±10% defined for this spaceborne sensor (JPSS, 2019) from the surface to ~10 km asl and above 10 km asl the CrIS IR-only retrievals exceeded this systematic bias requirement. Finally, the combined UV+IR retrievals consistently had NMB values lower than ±10% at all altitudes in the troposphere. The primary drivers of systematic biases were determined to be sza, surface albedo, and cloud fraction. The accuracy of all three retrievals tend to be degraded with increasing sza and cloud fraction and lower surface albedo values.

Just as important as systematic bias, this study validated the TROPOMI UV, CrIS IR, and TROPOMI/CrIS

UV+IR TOPAS retrievals for daily unresolved errors. Random error (uncertainty) requirements for TROPOMI UV
$O_3$ profile retrievals are $\pm 10\%$ (ESA, 2014) and $\pm 25\%$ for CrIS IR profiles in the troposphere (JPSS, 2019). The
validation of UV-only, IR-only, and UV+IR retrievals using the TOLNet-AK observations resulted in troposphere-
averaged RMSE values of 19.8%, 12.6%, and 14.6%, respectively. TROPOMI UV-only profiles evaluated here do
not meet the uncertainty requirements defined by the ESA. CrIS IR-only retrievals do meet the uncertainty

requirements defined by the mission. The ability of the retrievals to deviate from the a priori profile shape assumption
is key to lower systematic bias and unresolved error. UV-only retrievals have the least sensitivity to $O_3$ in the
troposphere leading to posterior vertical profiles with nearly identical shape compared to the prior (see Fig. 5). The
improved sensitivity of IR wavelengths to $O_3$ in the troposphere allows IR-only and UV+IR retrievals to deviate
further from the shape of the a priori resulting in lower systematic biases and unresolved errors (see Fig. 5 and Table

3).

The results of this validation study can be used to understand the biases and random errors associated with
TROPOMI UV, CrIS IR, and TROPOMI/CrIS UV+IR retrievals. While this study is specific to the TOPAS algorithm
it reflects the overall accuracy and precision of TROPOMI and CrIS $O_3$ vertical profile in the troposphere. The satellite
retrievals provide useful information for understanding tropospheric $O_3$; however, the sensitivity of TROPOMI UV-

only retrievals is still a limiting factor for accurately assessing variability in tropospheric $O_3$. IR-retrievals from CrIS
provide enhanced sensitivity to tropospheric $O_3$; however, it is limited by the coarse spatial resolution of the sensor
and lack of sensitivity to $O_3$ in the stratosphere. Combining these retrievals improves the ability to observe tropospheric
$O_3$ to some degree. Future work should consider building off the many studies which have retrieved tropospheric $O_3$
profiles with the combination of individual satellite retrievals with different wavelength ranges (e.g., Worden et al.,

2007b; Fu et al., 2013; Cuesta et al., 2013; Fu et al., 2018; Colombi et al., 2021; Mettig et al., 2022; Malina et al.,
2022) in order to continue improving the sensitivity of spaceborne retrievals of $O_3$ in the troposphere. TOLNet will
make dedicated observations for TEMPO validation to evaluate the combination of UV+VIS wavelengths. The
UV+VIS retrievals have enhanced lowermost tropospheric sensitivity, and in combination with the sensor's high
spatiotemporal resolution, should provide important spaceborne information of tropospheric column and lowermost

tropospheric $O_3$. While TEMPO $O_3$ profile and partial column data was not available at the time of this publication,
preliminary analysis suggests that the UV+VIS-derived 0-2 km partial column product from this geostationary sensor
should have DOF values between $0.2 - 0.3$ (Natraj et al., 2011; Zoogman et al., 2016; Johnson et al., 2018). An
important result of this study was showing that TOLNet is a sufficient validation data source for satellite retrievals
since TOLNet has been identified as the primary data source for validation of TEMPO $O_3$ in the troposphere.

*Data availability.* The TOLNet data used for the satellite data validation is available for download
(https://tolnet.larc.nasa.gov/, last access September 6, 2023). The TOPAS satellite retrievals are available upon request
to the corresponding author and University of Bremen coauthors. Ozonesonde data can be downloaded for GML
(https://gml.noaa.gov/dv/data/index.php?category=Ozone&type=Balloon&site=BLD, last access December 19,

2021), HUB (https://www-air.larc.nasa.gov/cgi-bin/ArcView/owlets.2018?SONDE=1, last access January 22, 2022),
HMI (https://www-air.larc.nasa.gov/cgi-bin/ArcView/owlets.2018?SONDE=1, last access January 22, 2022), UMBC
(https://www-air.larc.nasa.gov/cgi-bin/ArcView/owlets.2018?SONDE=1, last access January 22, 2022), FLP
(https://www-air.larc.nasa.gov/cgi-bin/ArcView/listos?GROUND-FLAX-POND=1, last access January 30, 2022),
WCT (https://www-air.larc.nasa.gov/cgi-bin/ArcView/listos?GROUND-WESTPORT=1, last access January 30,
2022), and RU (https://www-air.larc.nasa.gov/cgi-bin/ArcView/listos?GROUND-RUTGERS=1, last access January
30, 2022). Ozonesondes from UAH can be acquired by email to the corresponding author.

*Author contributions*. MSJ, JS, and MJN were responsible for acquiring the funding for this study. MSJ designed the
technical methods and performed the experiments. AR, MW, and NM developed and produced the satellite retrieval
data applied in this study. JS, MJN, SK, TL, FC, TAB, GG, RJA, AOL, CJS, GK, BC, and LT were instrumental in
obtaining and providing the lidar data used for validation. MSJ prepared the manuscript with contributions from all
coauthors.

*Competing interests*. At least one of the (co-)authors is a member of the editorial board of Atmospheric Measurement
Techniques.

*Disclaimer*. The views, opinions, and findings contained in this report are those of the authors and should not be
construed as an official NASA or United States Government position, policy, or decision.

*Acknowledgments*. The authors of this study acknowledge the financial support which made this study possible
(described below). Resources supporting this work were provided by the NASA High-End Computing (HEC) Program
through the NASA Advanced Supercomputing (NAS) Division at NASA Ames Research Center. Some calculations
were performed on high-performance computing (HPC) facilities of the IUP, University of Bremen, funded under
DFG/FUGG grant nos. INST 144/379-1 and INST 144/493-1. The authors gratefully acknowledge the computing time
granted by the Resource Allocation Board and provided on the supercomputer Lise at NHR@Berlin as part of the
national high performance computing infrastructure. The calculations for this research were conducted with computing
resources under the project hbp00072. GALAHAD Fortran Library was used for calculations. Retrievals use the
GALAHAD Fortran library. The S5P L1b version 2 data are provided by ESA/KNMI via the S5P validation team
(S5P-VT) activities. Level 1 and level 2 products from CrIS used in the retrieval are provided by NASA.

*Financial support*. MSJ, JS, MJN, SK, TL, FC, TAB, GG, RJA, AOL, CJS, GK, BC, and LT acknowledge the funding
from the NASA Tropospheric Composition Program through the TOLNet Science Team. The TROPOMI retrievals
at the University of Bremen and AR, MW, and NM were funded by the BMWi/German Aerospace Center (DLR)
project "S5P Datennutzung" (Förderkennzeichen 50EE1811A), the University of Bremen, and the federal state of
Bremen. KBS provided lidar retrievals through in-kind efforts.

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
