# Peer review of "TOLNet validation of satellite ozone profiles in the troposphere: impact of retrieval wavelengths"

_Atmospheric Measurement Techniques, 2023_

## Author Comment (AC1)

**Response to Reviewer #1**

We thank the reviewer for their detailed comments on the manuscript. We have addressed these comments as described below. All reviewer comments are presented in italic font while the author responses are displayed in standard font. Specific text that was added to the updated manuscript is provided in blue text.

*The study by Johnson et al., titled "TOLNet validation of satellite ozone profiles in the troposphere: impact of retrieval wavelengths" used lidar profiles of tropospheric ozone to evaluate the equivalent retrieved from TropOMI, CriS and TropOMI+CriS using the TOPAS algorithm. This represented retrieval schemes exploiting UV, IR and UV+IR wavelengths to retrieve tropospheric ozone. The long-term plan being to use TOLNet to evaluate tropospheric ozone profiles from the new TEMPO geostationary platform. Overall, this is a nice study demonstrating the suitability of this lidar network to evaluate satellite data, with the novel use of a larger network of lidars than previously used over the US. Therefore, this manuscript is suitable for publication in AMT subject to some minor comments below:*

*Page 3 Lines 80-88: The paragraph suggested that there are only two retrieval schemes of ozone profiles from OMI. However, the RAL Space retrieval scheme described by Miles et al., (2015) is used for GOME, GOME-2A & B, SCIAMACHY and OMI. Therefore, this should be mentioned in this paragraph and relevant references included (e.g. Keppens et al., (2018); Pope et al., (2020); Russo et al., 2023).*

Thank you for identifying this oversight. We have added the following text to the introduction section of the updated manuscript to identify and explain the RAL Space OMI algorithm: "There are three $O_3$ profile retrieval algorithms for OMI (NASA - Royal Netherlands Meteorological Institute (KNMI), van Oss et al., 2002; Smithsonian Astrophysical Observatory (SAO), Liu et al., 2010; Rutherford Appleton Laboratory (RAL) Space, Pope et al., 2023)" and "The RAL Space algorithm uses UV wavelengths (270–350 nm) to retrieve $O_3$ profiles at the native spatial resolution of the sensor (13 km × 24 km at nadir) with similar vertical resolution as the other two algorithms (Miles et al., 2015; Keppens et al., 2018; Pope et al., 2023).".

*Page 7 Line 196: Should the Jacobian matrix, K, be in bold?*

This has been corrected.

*Page 7 Line 205: Add "in" after "12 weeks" and before "total".*

This has been corrected.

*Equation 3: Is the more traditional method to write this equation as Xc = Xa + AK(Xt-Xa)? Also, I don't think Xc is defined.*

The $O_3$ profile TOPAS retrieval is conducted with relative deviations from the a priori as explained in Mettig et al. (2021) which is why we wrote Eq. (3) this way initially. However, to avoid confusion for the reader we have changed Eq. (3) as suggested by the reviewer. The following text has been added to the updated manuscript: "The TOPAS retrieval is conducted with relative

deviations from the $X_a$, therefore the **AK** is converted appropriately as explained in Mettig et al. (2021).”. We have also now defined *Xc* in the updated manuscript in the following sentence: “The satellite retrievals were compared to raw observations and when convolved ($X_c$) with the averaging kernel (**AK**) and a priori information from each retrieval using Eq. (3):”.

*Page 10 Figure 2: It is true that IR tends to have slightly more information on tropospheric ozone. However, I think one sentence discussing the total DOF (as you show it in your plot and provide numbers) would be useful as the UV scheme has much more sensitivity overall (though this is middle-upper atmosphere). E.g. add a sentence on Page 10 Line 279 outlining the general picture and then focus on the tropospheric component.*

We thank the reviewer for bringing up this good point. At the beginning of this paragraph, we have now added the sentence: “Each of the three retrievals display different total column DOFs (0-60 km asl) with UV+IR retrievals having the highest sensitivity (5.65) followed by UV-only (5.01) and IR-only (2.28).” along with other modifications to emphasize the differences in the retrieval’s DOFs.

*General point, the quality of the figures needs improving as many (especially the text) are pixelated.*

We appreciate this comment and will do our best to improve the quality of the figures before final publication, if approved by the reviewers and editor.

*In Figures 4,5,6 and 8, can the authors add a sentence making it clear what all the statistical metrics are (e.g. RMSE) and clearly state what the reference is. E.g. what you use as the reference to get the NMB numbers (e.g. apriori or TOLNET/ozonesondes convolved with the TOPAS Aks).*

This information has been added to the caption of Fig. 4, 5, 6, and 8 in the updated manuscript.

*Where possible, fit all of Table 2 onto a single page.*

The updated manuscript has been formatted so Table 2 and 3 are entirely on the same page.

*I find figure 8 slightly confusing. I can only see one TOLNet profile convolved by the AKs. However, as there are 3 retrievals for UV, IR and UV+IR, there should be 3 sets of AKs to convolve the TOLNet profiles. However, I don’t see this. Do the authors only use e.g. TOLNet + UV/IR AKs? And for the bias plots on the RHS, make it clear what the retrievals are compared to e.g. TOLNet + AKs from one retrieval or each wavelength retrieved compared with TOLNet + their corresponding AKs?*

Thank you for identifying this aspect of confusion. The reviewer is correct, on the left-hand side of Fig. 8 we only plot TOLNet convolved with the UV+IR AK. This is done to reduce the number of lines in the plot to avoid too much confusion for the reader. However, the NMB values calculated in the right-hand side of the figure use the respective AKs of each of the three retrievals. Figure 8 caption now reads as: “Seasonally-averaged vertical $O_3$ profile comparison of TOLNet interpolated to the satellite vertical grid (TOLNet-raw), TOLNet convolved with the TOPAS AKs (TOLNet-AK), UV, IR, and UV+IR TOPAS satellite retrievals, and the a priori profile information. The TOLNet profile convolved with the UV+IR AKs are displayed and the two other

(UV- and IR-only) convolved profiles are not shown to reduce the number of profiles presented. The direct comparison of the profiles (left column) and normalized mean bias (NMB) (right column) for UV-only, IR-only, and UV+IR retrievals compared to TOLNet-AK as the reference are displayed, respectively. NMB values for each of the three retrievals are calculated using the TOLNet profiles convolved with the correct retrieval-specific AK as the reference. NMB values of 30% and 10% are displayed using grey dashed and dotted lines, respectively. The total number (N) of co-located profiles are shown in the figure inset.".

*Page 24 Line 545: Why use TOLNet raw and not TOLnet+AKs?*

We thank the reviewer for catching this typo. We have corrected it to say "convolved TOLNet…".

*Page 24 Lines 549-550: Add some numbers for the RMSE stats discussed.*

We have expanded upon this claim presented in the revised paper to present the actual RMSE values calculated for the different observation characteristics (albedo, sza, and cloud fraction).

*Page 24 Lines 555-556: The statement "b) retrievals with minimal dependence on apriori information" is too strong in my opinion. If you were discussing only the tropospheric column, where Fig2 suggests the DOF is approximately 0.7-0.8, then I would be inclined to agree as you have nearly 1 piece of independent information from the troposphere. However, as you are looking at profiles, where the DOF will drop substantially, I would be inclined to replace "minimal dependence on apriori information" with "decent independence from the apriori information".*

We apologize for the confusion about this statement. Here we are discussing the capabilities of the TOLNet observations which have minimal dependance on a priori information (e.g., meteorological conditions). We have altered this statement as: "TOLNet data provides: a) highly accurate, high temporal resolution, $O_3$ observations for multiple continuous hours and/or days, b) retrievals with minimal dependance on a priori information, and c) profiles with higher vertical resolution compared to satellite products." in order to avoid this confusion.

*Page 25 Line 585: You discuss the sensitivity of the retrieved ozone to SZA, apriori and surface albedo, but would it be worth looking at cloud fraction? E.g. looking at a cloud fraction of 0.1 vs 0.2 on retrieved ozone? CF is an important factor in retrieving any quantity from space.*

We agree with the reviewer and have added this analysis to this section of the updated manuscript. The following text was added: "Cloud interference can impact retrievals of most atmospheric constituents such as $O_3$ profiles. Here it was determined that while systematic biases for low cloud scenes (cloud fraction < 0.2) and times of high clouds (cloud fraction > 0.2) were similar (~14 ppb), RMSE values were larger for cloudy scenes (17.1 ppb) compared to clear pixels (13.5 ppb). This study further emphasizes the impact that clouds can have a detrimental impact on the accuracy and uncertainties of $O_3$ profile retrievals.".

*Page 26 Line 609: "and lowermost tropospheric ozone." I'm not sure you can say that here as the DOF is low at 0.1. Please provide more justification for this statement.*

It has been suggested by previous studies that the TEMPO UV+VIS retrievals will have much larger sensitivity to lowermost tropospheric (0-2 km agl) $O_3$ compared to UV-only and UV+IR

retrievals. The following text has been added to the discussion section in the updated manuscript to support this statement: "While TEMPO O$_3$ profile and partial column data was not available at the time of this publication, preliminary analysis suggests that the UV+VIS-derived 0-2 km partial column product from this geostationary sensor should have DOF values between 0.2 – 0.3 (Natraj et al., 2011; Zoogman et al., 2016; Johnson et al., 2018).".

**References:**

Johnson, M. S., Liu, X., Zoogman, P., Sullivan, J., Newchurch, M. J., Kuang, S., Leblanc, T., and McGee, T.: Evaluation of potential sources of a priori ozone profiles for TEMPO tropospheric ozone retrievals, Atmos. Meas. Tech., 11, 3457–3477, https://doi.org/10.5194/amt-11-3457-2018, 2018.

Keppens, A., Lambert, J.-C., Granville, J., Hubert, D., Verhoelst, T., Compernolle, S., Latter, B., Kerridge, B., Siddans, R., Boynard, A., Hadji-Lazaro, J., Clerbaux, C., Wespes, C., Hurtmans, D. R., Coheur, P.-F., van Peet, J. C. A., van der A, R. J., Garane, K., Koukouli, M. E., Balis, D. S., Delcloo, A., Kivi, R., Stübi, R., Godin-Beekmann, S., Van Roozendael, M., and Zehner, C.: Quality assessment of the Ozone_cci Climate Research Data Package (release 2017) – Part 2: Ground-based validation of nadir ozone profile data products, Atmos. Meas. Tech., 11, 3769–3800, https://doi.org/10.5194/amt-11-3769-2018, 2018.

Mettig, N., Weber, M., Rozanov, A., Arosio, C., Burrows, J. P., Veefkind, P., Thompson, A. M., Querel, R., Leblanc, T., Godin-Beekmann, S., Kivi, R., and Tully, M. B.: Ozone profile retrieval from nadir TROPOMI measurements in the UV range, Atmos. Meas. Tech., 14, 6057–6082, https://doi.org/10.5194/amt-14-6057-2021, 2021.

Miles, G. M., Siddans, R., Kerridge, B. J., Latter, B. G., and Richards, N. A. D.: Tropospheric ozone and ozone profiles retrieved from GOME-2 and their validation, Atmos. Meas. Tech., 8, 385–398, https://doi.org/10.5194/amt-8-385-2015, 2015.

Natraj, V., Liu, X., Kulawik, S. S., Chance, K., Chatfield, R., Edwards, D. P., Eldering, A., Francis, G., Kurosu, T., 790 Pickering, K., Spurr, R., and Worden, H.: Multispectral sensitivity studies for the retrieval of tropospheric and lowermost tropospheric ozone from simulated clear sky GEO-CAPE measurements, Atmos. Environ., 45, 7151–7165, https://doi.org/10.1016/j.atmosenv.2011.09.014, 2011.

Pope, R. J., Kerridge, B. J., Siddans, R., Latter, B. G., Chipperfield, M. P., Feng, W., Pimlott, M. A., Dhomse, S. S., Retscher, C., and Rigby, R.: Investigation of spatial and temporal variability in lower tropospheric ozone from RAL Space UV–Vis satellite products, Atmos. Chem. Phys., 23, 14933–14947, https://doi.org/10.5194/acp-23-14933-2023, 2023.

Zoogman, P., Liu, X., Suleiman, R. M., Pennington, W. F., Flittner, D. E., Al-Saadi, J. A., Hilton, B. B., Nicks, D. K., Newchurch, M. J., Carr, J. L., Janz, S. J., Andraschko, M. R., Arola, A., Baker, B. D., Canova, B. P., Chan Miller, C., Cohen, R. C., Davis, J. E., Dussault, M. E., Edwards, D. P., Fishman, J., Ghulam, A., González Abad, G., Grutter, M., Herman, J. R., Houck, J., Jacob, D. J., Joiner, J., Kerridge, B. J., Kim, J., Krotkov, N. A., Lamsal, L., Li, C., Lindfors, A., Martin, R. V., McElroy, C. T., McLinden, C., Natraj, V., Neil, D. O., Nowlan, C. R., O'Sullivan, E. J., Palmer, P. I., Pierce, R. B., Pippin, M. R., Saiz-Lopez, A., Spurr, R. J. D., Szykman, J. J., Torres, O., Veefkind, J. P., Veihelmann, B., Wang, H., Wang, J., and Chance, K.: Tropospheric emissions: Monitoring of pollution (TEMPO), J. Quant. Spectrosc. Ra., 186, 17–39, https://doi.org/10.1016/j.jqsrt.2016.05.008, 2016.

---

## Author Comment (AC2)

**Response to Reviewer #2**

We thank the reviewer for their detailed comments on the manuscript. We have addressed these comments as described below. All reviewer comments are presented in italic font while the author responses are displayed in standard font. Specific text that was added to the updated manuscript is provided in blue text.

*General comments:*

*The paper is intended to describe validation of UV, IR, and UV+IR ozone retrievals based on TROPOMI and CRIS data and the TOPAS retrieval algorithm. The validation data sets are TOLNET (lidar) and ozone-sondes. I have three main concerns with this paper. Firstly, it is not clear how this paper contributes to our understanding of the TOPAS CRIS/TROPOMI ozone retrievals over a previous paper (essentially from the same group) by Mettig et al. (AMT 2022) as well as un-cited work by Malina et al (AMTD). The paper indicates that they use the "full capabilities" of the TOLNET data sets but I could not find what this means or how it advances the validation of these retrievals relative to what is describe in Mettig et al. The paper suggests that a key result is that using UV+IR radiances improves the ozone retrievals over use of UV or IR radiances alone; however this result is already well known and well described in other papers (some cited, some not). My second main concern is that while the paper is well organized, it is poorly written with numerous non-quantitative statements, and with essentially no context, introduction, or comparisons to other work within most of the results component of the paper. Thirdly, much of the paper appears to be repetitive with respect to the Mettig et al. paper.*

*To get through review, I would ask that the authors spend more effort in describing what is new and different about this paper relative to the Mettig et al. paper; likely this would also help in shortening the paper as material that appears in Mettig et al. need not be restated in the submitted manuscript. Adding context to each of the results sub sections and how what is presented is similar/different to previous work would also improve the writing.*

We thank the reviewer for their suggestions on how to improve our manuscript. We have made substantial changes to the text in the revised manuscript to address these concerns. The main improvements are listed below:

- Numerous sections in the updated manuscript have been revised to compare the results from our study to the two other TROPOMI/CrIS O$_3$ profile validation studies from Mettig et al. (2022) and Malina et al. (2022). Furthermore, the comparison to other multi-wavelength O$_3$ vertical profile retrievals, using different satellite sensors, is now discussed and the appropriate references are provided (e.g., Landgraf and Hasekamp, 2007; Worden et al., 2007b; Cuesta et al., 2013, 2018; Costantino et al., 2017; Colombi et al., 2021; Malina et al., 2022; Mettig et al., 2022). For example, in Sect. 4 of the revised manuscript our results are compared to Cuesta et al. (2013) which used UV+IR retrievals from GOME-2/IASI, which was not done in detail in Mettig et al. (2022): "Applying different combinations of UV+IR joint wavelength retrievals (e.g., GOME-2+IASI) also displays improvements compared to UV-only products in the troposphere similar to that determined in this study (e.g., Cuesta et al., 2013, 2018). Cuesta et al. (2013, 2018) demonstrated how GOME-2+IASI retrievals show high accuracy compared to ozonesondes in the lowermost troposphere and displays a clear capability to capture PBL

O₃ enhancements. This differs from the results of this study which suggest that TROPOMI+CrIS UV+IR joint wavelength retrievals still struggle to reproduce large PBL O₃ enhancements due to limited lowermost tropospheric sensitivity. The reasons why GOME-2+IASI displays the remarkable capability to retrieve lowermost tropospheric enhancements compared to the results from TROPOMI+CrIS is not immediately apparent. There are differences in the retrieval algorithms, a priori input data sets, and the spectral resolutions of the UV and IR sensors applied. Comparing our results to Cuesta et al. (2013) shows that DOFs are higher in the troposphere and in the 0-2 km agl column (>33% higher) in GOME-2+IASI retrievals compared to TROPOMI+CrIS which would explain some of the differences in capabilities to retrieve lowermost tropospheric O₃ enhancements.".

- Multiple sections of the revised manuscript were updated to explain how our study expands upon Mettig et al. (2022) and Malina et al. (2022). In the introduction we state: "This study builds upon Mettig et al. (2022) to demonstrate the full capability of TOLNet (6 of the 8 systems that were available for the first year of TROPOMI observations) to validate satellite O₃ retrievals at multiple vertical levels in the troposphere. This study applies all available TOLNet systems with spatial coverage throughout the US and in the Netherlands, compared to the small subset of 2 lidar systems used in Mettig et al. (2022), to conduct a more robust validation of the UV-only TROPOMI, IR-only CrIS, and UV+TIR TROPOMI/CrIS O₃ profile retrievals. Furthermore, the only other study to validate TROPOMI/CrIS UV+IR retrievals (Malina et al., 2022) did not apply any ground-based lidar observations. Finally, this study conducts a detailed statistical analysis of satellite O₃ profile retrievals at various vertical levels of the troposphere and investigates the capability of these retrievals to reproduce anomalous atmospheric composition with large impacts on air quality (e.g., stratospheric intrusions, lowermost troposphere pollution events) which was not conducted in past studies validating TROPOMI/CrIS UV+IR retrievals in the troposphere (Mettig et al., 2022; Malina et al., 2022).". Demonstrating the capability of TOLNet to be used as a satellite O₃ validation data set has not yet been proven in the literature and is a major objective of this study. The importance of this is expanded upon in the introduction: "Demonstrating the capability of TOLNet to sufficiently validate satellite O₃ profiles is vital as TOLNet is the primary validation data source for validating TEMPO O₃ products in the troposphere. To-date, no studies have validated satellite data with TOLNet beyond 1 or 2 individual systems instead of the entire network (8 total lidar systems) (Mettig et al., 2022; Sullivan et al., 2022).".

- Our study now demonstrates the similarities and differences between the validation conducted here and the results from Mettig et al. (2022) and Malina et al. (2022). Even the similarities such as demonstrating the improvement in tropospheric O₃ profile retrievals when using UV+IR wavelengths compared to UV-only, while it has been shown in past studies, is important to prove that TOLNet can be used to validate satellite data as accurately as those that applied ozonesondes. We attempted to emphasize this objective in the original manuscript but have added additional text to help highlight this point such as that implemented into the results section of the updated manuscript: "The agreement in the validation statistics of TROPOMI UV, CrIS IR, and TROPOMI/CrIS UV+IR retrievals determined in this study when using TOLNet-AK and those using primarily ozonesonde data (Mettig et al., 2022; Malina et al., 2022) demonstrates that TOLNet is a sufficient validation source for satellite O₃ profile retrievals in the troposphere." and "It is important to note that TOLNet and ozonesonde validation statistics are generally consistent given the fact that ozonesondes are a highlyaccurate and commonly-applied satellite validation data source. This suggests that TOLNet is a sufficient validation data source of tropospheric $O_3$ profile retrievals from satellites.".

- Besides using all the available TOLNet systems, which was not done in either Mettig et al. (2022) and Malina et al. (2022), we focus on chemical environments which are critical for air quality and tropospheric composition which can be challenging to retrieve from space (i.e., stratospheric intrusions, PBL $O_3$ enhancements) which was not done in Mettig et al. (2022) and Malina et al. (2022). This is emphasized in the updated manuscript: "This analysis of complex atmospheric environments important for air quality using idealized retrievals, produced with known $O_3$ profiles convolved separately with different retrieval AKs, in this study expands on past studies that have evaluated TROPOMI/CrIS retrievals (Mettig et al., 2022; Malina et al., 2022). It is important to understand the extent to which TROPOMI, CrIS, and TROPOMI/CrIS joint satellite retrievals, which rely on different wavelengths, can accurately retrieve typical and anomalous structures of $O_3$ in the troposphere.". In addition to this, our study conducts a very detailed validation of satellite $O_3$ profiles at multiple vertical levels of the troposphere which was not done in Mettig et al. (2022) and Malina et al. (2022). This is now emphasized in the updated manuscript: "This TROPOMI/CrIS validation at multiple layers in the troposphere allows for more detailed interpretation of the capability of these satellite vertical profiles to retrieve middle- to lower-tropospheric $O_3$ in comparison to other recent TROPOMI/CrIS validation studies (Malina et al., 2022; Mettig et al., 2022).".

- The reviewer states that many non-quantitative statements were made in the original manuscript. No examples were given so it was not immediately clear what they were referring to. However, we have gone through and added quantitative information in many sections of the updated manuscript to address this reviewer comment.

*I next have a few specific comments just for the abstract and more general comments / questions about the paper thereafter.*

*Abstract:*

*(first paragraph) It is already well known that use UV+IR radiances to estimate ozone increases sensitivity (vertical resolution), relative to UV and IR alone; it is therefore not clear why this first paragraph in the abstract is needed.*

This first paragraph was shortened slightly to remove the opening statement about the increased sensitivity in UV+IR retrievals compared to UV- and IR- only retrievals. However, the quantitative statement about the degree of increase in the troposphere, and the improved ability to retrieve high $O_3$ conditions in the upper/mid and lowermost troposphere, were retained.

*Line 40: What are the "tropospheric systematic bias requirements"? Is there a source?*

This statement in the abstract was updated to read "…meet the tropospheric systematic bias requirements defined by the science teams for the TROPOMI and CrIS sensors".

*Line 41: If the averaging kernel and prior (observation operator) were applied to the TOLNET profiles before comparison than the a priori is removed from the comparison; therefore this should*

*not be a source of systematic bias unless you can show that non-linearities in the inversion make the choice of prior affect the inversion.*

The reviewer is correct. The bias in the magnitudes and shape of the a priori $O_3$ profile only biases the retrievals in comparison to raw observations. The statements about the a priori bias impacts when compared to convolved observations have been removed from the updated manuscript and this sentence now reads: "The primary drivers of systematic bias were determined to be solar zenith angle, surface albedo, and cloud fraction.".

*Line 47: "random bias"? Please clarify.*

This sentence has been updated in the revised manuscript to: "Random errors, representative of uncertainty in the retrievals and quantified by root mean squared errors (RMSE),…" to reflect how random errors are quantified.

*Line 52: If TOLNET was sufficient why also use ozonesonde data. Also what does sufficient mean?*

In order to show that TOLNet was sufficient, or has the capability to be used, for validating satellite $O_3$ profile retrievals in the troposphere, besides the fact these lidar data have been validated in past research and shown to be highly accurate as discussed in the manuscript, it is important to see whether TOLNet results in similar validation statistics compared to the well-known satellite validation data source from ozonesondes. The final sentence of the abstract has been updated to read: "TOLNet was shown to result in similar validation statistics compared to ozonesonde data, which are a commonly-used satellite evaluation data source, demonstrating that TOLNet is a sufficient source of satellite $O_3$ profile validation data in the troposphere which is critical as this data source is the primary product identified for the tropospheric $O_3$ validation of the recently-launched Tropospheric Emissions: Monitoring of Pollution (TEMPO) mission.". The similarities in validation results determined in this study, compared to other past TROPOMI/CrIS validation studies, which primarily used ozonesondes, are important to demonstrate the capabilities of TOLNet to validate a satellite product. As noted above, we attempted to emphasize this objective in the original manuscript but have added additional text to help highlight this point such as that implemented into the results section of the updated manuscript: "The agreement in the validation statistics of TROPOMI UV, CrIS IR, and TROPOMI/CrIS UV+IR retrievals determined in this study when using TOLNet-AK and those using primarily ozonesonde data (Mettig et al., 2022; Malina et al., 2022) demonstrates that TOLNet is a sufficient validation source for satellite $O_3$ profile retrievals in the troposphere." and "It is important to note that TOLNet and ozonesonde validation statistics are generally consistent given the fact that ozonesondes are a highly-accurate and commonly-applied satellite validation data source. This suggests that TOLNet is a sufficient validation data source of tropospheric $O_3$ profile retrievals from satellites.".

*Other comments*

*There are far more ozone-sondes available than just the ones listed in Table 1. Why do you not use them?*

We use a small subset of ozonesonde data as it is important to intercompare the validation statistics from TOLNet and ozonesondes separately (as mentioned above this was a goal of this study). In order to do this, we must use observations from both sources taken close to the same locations and time periods. This is further explained in Sect. 2.2 of the revised manuscript with the following statement: "Ozonesondes have been used extensively to validate satellite $O_3$ vertical profiles in past research (e.g., Worden et al., 2007a; Kroon et al., 2011; Verstraeten et al., 2013; Huang et al., 2017; Malina et al., 2022). In addition to the fact that TOLNet lidar data has been shown to be highly accurate (Leblanc et al., 2016, 2018) as discussed above, this study intercompares the validation statistics from spatially and temporally collocated TOLNet and ozonesonde observations to demonstrate the capability of TOLNet to be used for validating satellite $O_3$ vertical profiles.".

*Equations 1 and 2 are inconsistent with Equation 3. Equations 1 and 2 indicate that the retrieval is linear with respect to concentration or VMR. Equation 3 suggests either a log or fractional value is estimated; additional explanation is required.*

In accordance with a comment from Reviewer #1, the way we present Eq. (3) has been corrected in the revised manuscript. Furthermore, the following sentence was added for explanation: "The TOPAS retrieval is conducted with relative deviations from the $X_a$, therefore the $AK$ is converted appropriately as explained in Mettig et al. (2021)".

*There is another paper on this subject by Edward Malina that is not cited. The authors should take a look at this paper and describe what is different with their approach and results relative to those in Malina et al.*

*Joint spectral retrievals of ozone with Suomi NPP CrIS augmented by S5P/TROPOM*

*Malina, E., Bowman, K. W., Kantchev, V., Kuai, L., Kurosu, T. P., Miyazaki, K., Natraj, V., Osterman, G. B., and Thill, M. D.: Joint spectral retrievals of ozone with Suomi NPP CrIS augmented by S5P/TROPOMI, EGUsphere [preprint], https://doi.org/10.5194/egusphere-2022-774, 2022.*

We thank the reviewer for pointing us to this paper. We now reference the work by Malina et al. (2022) and compare our results to this study throughout the revised manuscript. In addition to the comparison of our results to Mettig et al. (2020) and Malina et al. (2022), the following text was added to summarize the similarities and differences of the approaches of the studies of Mettig et al. (2022) and Malina et al. (2022): "Multiple recent studies have combined UV+IR wavelength retrievals from two newer satellite sensors TROPOMI and CrIS to retrieve tropospheric $O_3$ vertical profiles (Mettig et al., 2022; Malina et al., 2022). The combined UV+IR TROPOMI/CrIS $O_3$ profile retrievals from Mettig et al. (2022) were evaluated in the troposphere for a full-year between 2018-2019 using a small sample (2 lidar systems which are also part of the Tropospheric Ozone Lidar Network (TOLNet)) of ground-based lidar remote-sensing observations from the Network for the Detection of Atmospheric Composition Change (NDACC) and ozonesondes (i.e., World Ozone and Ultraviolet Radiation Data Center (WOUDC) and the Southern Hemisphere Additional Ozonesondes (SHADOZ)) and demonstrated that the combined UV+IR retrievals were more consistent with observations compared to the UV-only product. Malina et al. (2022) also

evaluated a full-year (between 2019-2020) of combined UV+IR TROPOMI/CrIS $O_3$ profiles using correlative satellite retrievals and ozonesondes and further showed that combined UV+IR retrievals were more accurate in the troposphere compared to UV- and IR-only products. Mettig et al. (2022) and Malina et al. (2022) both combined TROPOMI and CrIS retrievals; however, applied different retrieval algorithms, a priori input data, and portions of the spectral bands from each satellite, thus the validation results differed to some degree which is discussed in the current manuscript.".

*Line 105.. missing Worden et al. GRL 2007 reference where this is first discussed.*

This reference has been added.

*Section 3: Combinations of UV and IR have appeared in several papers over the last decade. How do the results appearing in each sub-section compare to this prior research (answer, you are essentially getting what is expected based on this prior research).*

We agree with the reviewer and have done our best to compare our results to past research throughout the revised manuscript.

*Figure 4: Are the UV, IR, and UV+IR, retrievals consistent (especially Figure 4c). Use Rodgers and Connor 2003 (not cited) to determine if purple, red, and blue are consistent or if differences in the troposphere are driven by attributable systematic errors (e.g. albedo, clouds) or because there is a lack of sensitivity in the troposphere.*

I believe there was some confusion about what is presented in Fig. 4 and discussed in Sect. 3.2 of the original manuscript. This figure shows example TOPAS retrievals produced with TOLNet lidar profiles convolved with AKs and a priori profiles from UV-only (blue), IR-only (red), and UV+IR (purple) retrieval information. The figure caption of Fig. 4 has been updated to read "Example TOPAS retrievals produced from TOLNet lidar profiles convolved with AKs and a priori profiles from UV-only (blue), IR-only (red), and UV+IR (purple) retrievals at the location of RO$_3$QET (34.73 °N, 86.65 °W) from the surface to 40 km asl for the case studies of: a) clean/background conditions, b) PBL pollution enhancement, and c) stratospheric intrusion." to better clarify this point. The legend in this figure also now states "True convolved -" instead of "Retrieval" in the updated version of the manuscript. Finally, to avoid any other confusion about this, we have made other small changes to Sect. 3.2 of the updated version of the paper and refer the reader back to Sect. 2.4 where we describe how example retrievals are calculated.

The AKs and error covariance matrices for each TOPAS retrieval are produced with a radiative transfer model that only accounts for known uncertainties (i.e., noise in the retrieval) and do not reflect the impact of clouds/aerosol/albedo/etc. Therefore, all the differences seen in Fig. 4 between the three retrievals are driven by the differences in sensitivity to $O_3$ in the troposphere.

*Line 340: where are these requirement thresholds described?*

The accuracy requirements for CrIS are described in Table 5.2.8 of the referenced document in the original manuscript (JPSS, 2019). The link to the supplemental portion of this document has been

corrected in the revised manuscript (and below in the Reference section). The table was misunderstood when we referenced the lower ($\pm 10\%$) and higher ($\pm 20\%$) bias requirements. The statements about the higher bias requirements have been removed from the revised manuscript.

*Line 395: This approach makes no sense.. if you do not apply the observation operator, then there will be a bias from the combination of prior and sensitivity.*

We assume the reviewer is referring to the paragraph between Line 396-404 in the original manuscript. We compare observations (i.e., TOLNet and ozonesondes) convolved with retrieval AKs and a priori profiles (AK-convolved) and the actual observations (raw). We focus the validation of the satellites using the AK-convolved observations which is well-described in the manuscript. A similar approach of comparing TROPOMI/CrIS UV+IR satellite retrievals against ozonesondes without the satellite observation operator being applied was also conducted in Malina et al. (2022) and Mettig et al. (2022). The comparison of satellite data to raw observations in the troposphere is important as this allows for the understanding of how the satellite retrievals are able to replicate actual $O_3$ values, not just the capability of the spaceborne sensors. This has been emphasized with the additional statement added to the updated manuscript: "While observations convolved with the observation operator is the primary validation data source, comparing the three retrievals to TOLNet observations not convolved with the retrieval AKs (hereinafter TOLNet-raw) is also important to understand how the satellite retrievals reproduce actual atmospheric composition in the troposphere.".

*Line 427: This is an interesting statement about TOLNET "A major advantage of using TOLNet for validation of satellite O3 profile retrievals is the ability to make observations at different vertical levels of the troposphere over an entire day or more. " However, it is not obvious how this capability is used for the comparisons.*

The ability of TOLNet to make high vertical and temporal resolution $O_3$ observations through the troposphere for many consecutive hours (TOLNet can make multi-day continuous observations) is a unique feature of these lidar systems. This aspect of TOLNet will be vital for validation of TEMPO, and other future geostationary sensors over the United States, retrievals of hourly $O_3$ profiles throughout the entire day. To better clarify this and emphasize the importance of TOLNet observations for this study, we added the following text to this section of the revised manuscript: "While TROPOMI and CrIS are polar-orbiting systems which only retrieve $O_3$ profiles once per day, the observations throughout an entire day are vital for validating geostationary sensors such as TEMPO. However, the high vertical resolution and accurate $O_3$ observations from TOLNet are applied here to robustly validate satellite retrievals at multiple layers of the troposphere.". Our study conducted a very detailed validation of satellite $O_3$ profiles at multiple vertical levels of the troposphere which was not done in Mettig et al. (2022) and Malina et al. (2022). This is now emphasized in the updated manuscript: "This TROPOMI/CrIS validation at multiple layers in the troposphere allows for more detailed interpretation of the capability of these satellite vertical profiles to retrieve middle- to lower-tropospheric $O_3$ in comparison to other recent TROPOMI/CrIS validation studies (Malina et al., 2022; Mettig et al., 2022).".

*Section 3.3.3. Again, what is different about these comparisons and conclusions versus Mettig et al. and Malina et al.?*

As mentioned in response to earlier comments, we now have included more comparison to our results and those from Malina et al. (2022) and Mettig et al. (2022) throughout the revised manuscript.

**References:**

[revised manuscript text omitted]

---

## Author Comment (AC3)

**Response to Reviewer #3**

We thank the reviewer for their detailed comments on the manuscript. We have addressed these comments as described below. All reviewer comments are presented in italic font while the author responses are displayed in standard font. Specific text that was added to the updated manuscript is provided in blue text.

*The manuscript submitted by Johnson and colleagues is a follow up of the work carried out by Mettig et al., 2022 using tropospheric ozone profiles reconstructed with the TOPAZ tool developed by the University of Bremen to exploit the synergy of UV (TROPOMI) and IR (CrIS) satellite observations. In this work the comparison is made over an 18-month period of TOPAZ retrieval and ground-based observations in North America (TOLNET lidar network and ECC ozonesondes). Mettig has already discussed the extent to which the synergy between UV and IR can improve the restitution of tropospheric ozone profiles but with a different validation data set based on NDACC observations in Europe and the USA. In the present work, the sensitivity study shown in Fig. 4 and the analysis of differences in several tropospheric layers are very useful, and was not present in that of Mettig et al. This work therefore deserves to be published in AMT, especially with the prospect of using TOLNET to validate the future GEO-TEMPO satellite mission.*

*My only minor concerns, which should be studied even if not taken into account, are the followings:*

*1) The discussion is sometimes based on the use of ground data convolved with AK of TOPAZ and sometimes based on the raw data interpolated vertically. It is better to use always the same criteria for the comparison of the three configurations. Use of the raw data should be made only for a better understanding of the results.*

We thank the reviewer for this comment. For the satellite validation we compare observations (i.e., TOLNet and ozonesondes) convolved with retrieval AKs and a priori profiles (AK-convolved). Overall, we focus the validation of the satellites using the AK-convolved observations which we feel is well-described in the manuscript. However, it is also important to understand how satellite retrievals are able to replicate actual $O_3$ values, not just the capability of the spaceborne sensors, which was also done in other TROPOMI/CrIS validation studies (Mettig et al., 2022; Malina et al., 2022). To emphasize this, we have added the following text to the first section of the revised manuscript using TOLNet-raw data: "While observations convolved with the observation operator is the primary validation data source, comparing the three retrievals to TOLNet observations not convolved with the retrieval AKs (hereinafter TOLNet-raw) is also important to understand how the satellite retrievals reproduce actual atmospheric composition in the troposphere.".

*2) The improvement when using the UV+IR configuration compared with IR-only is real for certain altitude ranges (boundary layer, UTLS) and for certain types of ozone profile (stratospheric intrusion), but does not significantly improve IR-only performance for other cases. This is not sufficiently recognized in the discussions of Fig. 5-6 and tables 2-3.*

Sect. 3.3.3 of the revised manuscript, which focuses on the comparison of the 3 retrievals at multiple tropospheric layers, has been updated significantly to better describe the performance of IR-only retrievals in comparison to the two other retrievals. We also provide more quantitative

information about the 3 retrievals evaluation at each vertical layer. At many points in this section, we now show how IR-only retrievals actually perform better compared to UV-only and UV+IR retrievals. An example of how we demonstrate this point is as follows: "Overall, between 2-8 km asl, IR-only retrievals have the least bias and spread, along with best linear regression fits. UV+IR retrievals are similar to IR-only data with only slightly worse performance when compared to TOLNet-AK. This result demonstrates that while the combination of UV and IR wavelengths tends to improve the performance of TOPAS retrievals compared to UV-only, this is not always the case for IR-only.".

*3) It's a pity that the ozonesonde measurements are not used in conjunction with those from TOLNET for the scatterplots shown in each altitude layers (Fig. 7) and for the analysis of the seasonal variability (Fig.8). This would increase the representativeness of the results, as ozone distributions from TOLNET and ozonesondes are clearly complementary. We are left with the impression that the ozonesonde data have been discarded in the second part of the paper because they do not show a decisive contribution from IR+UV compared with IR-only in Fig. 6 and Table 3.*

We appreciate this comment. However, the focus of this study is to demonstrate the network-wide TOLNet capability to validate satellite $O_3$ retrievals. In order to show that TOLNet was sufficient for validating satellite $O_3$ profile retrievals in the troposphere, besides the fact these lidar data have been evaluated in past research and are shown to be highly accurate as discussed in the manuscript, it is important to see whether TOLNet results in similar validation statistics compared to the well-known satellite validation data source from ozonesondes. The final sentence of the abstract has been updated to read: "TOLNet was shown to result in similar validation statistics compared to ozonesonde data, which are a commonly-used satellite evaluation data source, demonstrating that TOLNet is a sufficient source of satellite $O_3$ profile validation data in the troposphere which is critical as this data source is the primary product identified for the tropospheric $O_3$ validation of the recently-launched Tropospheric Emissions: Monitoring of Pollution (TEMPO) mission.". If we combine the two validation data sources (i.e., TOLNet and ozonesondes) it is not possible to determine the similarities and differences between the validation using TOLNet and the well-known validation data source from ozonesones. The similarities in validation results determined in this study, compared to other past TROPOMI/CrIS validation studies (e.g., Mettig et al., 2022; Malina et al., 2022), which primarily used ozonesondes, are also important to demonstrate the capabilities of TOLNet to validate a satellite product. As noted above, we attempted to emphasize this objective in the original manuscript but have added additional text to help highlight this point such as that implemented into the results section of the updated manuscript: "The agreement in the validation statistics of TROPOMI UV, CrIS IR, and TROPOMI/CrIS UV+IR retrievals determined in this study when using TOLNet-AK and those using primarily ozonesonde data (Mettig et al., 2022; Malina et al., 2022) demonstrates that TOLNet is a sufficient validation source for satellite $O_3$ profile retrievals in the troposphere." and "It is important to note that TOLNet and ozonesonde validation statistics are generally consistent given the fact that ozonesondes are a highly-accurate and commonly-applied satellite validation data source. This suggests that TOLNet is a sufficient validation data source of tropospheric $O_3$ profile retrievals from satellites.".

*Detailed questions or suggestions*

*Abstract line 27: Since contrary to Mettig, 2022 data in Europe are very limited in this work (10 % of the data base in September 2019), it is better to replace « Europe » by « Netherland in September 2019 »*

We agree with the reviewer, and this has been corrected in the abstract of the revised manuscript.

*Abstract line 51: TOLNET data are certainly consistent for a seasonal analysis, is it also true for the altitude range analysis?*

We apologize for the confusion this text caused. We have removed "Consistent daily" from the beginning of the sentence to remove any potential confusion. We did not want to suggest that the $O_3$ profiles from TOLNet were consistent within seasons or altitude ranges. We were attempting to state that the lidars can consistently provide data; however, this is not important to the results of this study, so we revised the sentence.

*Line 104: The introduction provides a very nice review of the different satellite missions including their horizontal and vertical resolution. A table to summarize these resolutions would be useful.*

This information has been implemented in the updated manuscript as Table 1.

*Line 112: Mettig et al. study also includes NDACC and SHADOZ observations in Europe and the Tropics (ozonesonde and lidar) in addition to the TOLNET lidar in California and Huntsville. The sentence should be changed to mention it.*

The sentence has been modified in the revised manuscript to read: "The combined UV+IR TROPOMI/CrIS $O_3$ profile retrievals from Mettig et al. (2022) were evaluated in the troposphere for a full-year between 2018-2019 using a small sample (2 lidar systems which are also part of the Tropospheric Ozone Lidar Network (TOLNet)) of ground-based lidar remote-sensing observations from the Network for the Detection of Atmospheric Composition Change (NDACC) and ozonesondes (i.e., World Ozone and Ultraviolet Radiation Data Center (WOUDC) and the Southern Hemisphere Additional Ozonesondes (SHADOZ)) and demonstrated that the combined UV+IR retrievals were more consistent with observations compared to the UV-only product.".

*Line 125: In order to clarify the contribution of this new study in relation to the work of Mettig et al., might be good to add « with an emphasis on North America and many lidar instruments» after « O3 profile retrieval ». It might be good to specify here that a detailed statistical analysis at different altitude ranges is conducted in this work while this point was not developed in Mettig et al.*

We agree with the reviewer that a more direct statement here would help separate this work from Mettig et al. (2022) and the previously uncited work by Malina et al. (2022). Besides using all the available TOLNet systems, which was not done in either Mettig et al. (2022) and Malina et al. (2022), we focus on chemical environments which are critical for air quality and tropospheric composition which can be challenging to retrieve from space (i.e., stratospheric intrusions, PBL $O_3$ enhancements) which was not done in Mettig et al. (2022) and Malina et al. (2022). This is emphasized in the updated manuscript: "This analysis of complex atmospheric environments

important for air quality using idealized retrievals, produced with known $O_3$ profiles convolved separately with different retrieval AKs, in this study expands on past studies that have evaluated TROPOMI/CrIS retrievals (Mettig et al., 2022; Malina et al., 2022). It is important to understand the extent to which TROPOMI, CrIS, and TROPOMI/CrIS joint satellite retrievals, which rely on different wavelengths, can accurately retrieve typical and anomalous structures of $O_3$ in the troposphere.". In addition to this, our study conducts a very detailed validation of satellite $O_3$ profiles at multiple vertical levels of the troposphere which was not done in Mettig et al. (2022) and Malina et al. (2022). This is now emphasized in the updated manuscript: "This TROPOMI/CrIS validation at multiple layers in the troposphere allows for more detailed interpretation of the capability of these satellite vertical profiles to retrieve middle- to lower-tropospheric $O_3$ in comparison to other recent TROPOMI/CrIS validation studies (Malina et al., 2022; Mettig et al., 2022).". Furthermore, numerous sections in the updated manuscript have been revised to compare the results from our study to the two other TROPOMI/CrIS $O_3$ profile validation studies from Mettig et al. (2022) and Malina et al. (2022).

*Line 145: The number of lidar observations considered (185) is different from the maximum value in table 2 (176). They should be consistent. The Mettig et al. study is finally not so different (170 lidar data and 200 ozonesondes for the same time period 2018/2019)*

Table 2 in the revised manuscript presents the total number of days with observations for each lidar system and location. This sums to 185 which is what is stated in Line 145 of the original manuscript. Due to limitations of lidar retrievals due to inability to retrieve $O_3$ values accurately in high cloud and/or aerosol conditions, and saturation due to solar background becoming too large and saturating the lidar signal, the numbers in Table 3 of the revised manuscript will differ between vertical levels. Also, the TOPAS retrieval provides data at 1 km, thus more than one satellite/TOLNet co-location for the 2 km bins for each of the 89 total profile co-locations (N = 89 in Fig. 5) is possible.

The reason that the number of lidar profiles used in our study is similar to that in Mettig et al. (2022), even though we use more lidar systems, is that we use stricter colocation criteria. This is explained in the original manuscript starting in Line 238. We have updated this text in the revised manuscript to be more specific: "Statistical comparisons between co-located satellite retrievals and observations were conducted using spatiotemporal thresholds of 2.5 hours and 30 km. Sensitivity studies were conducted using coarser co-location spatiotemporal thresholds of 5 hours and 100 km to maximize the number of co-locations for statistical evaluation and to be more consistent with recent TROPOMI/CrIS $O_3$ profile validation studies which use looser colocation thresholds (Mettig et al., 2021, 2022). As this study focuses on tropospheric $O_3$ which has large spatiotemporal variability, we feel the stricter spatiotemporal thresholds are most appropriate.".

*Line 165: give here the seasonal distribution of the TOLNET observations given line 473*

The following sentence has been added to this section of the revised manuscript: "This study includes 13, 28, 78, and 66 TOLNet observations for the winter (DJF), spring (MAM), summer (JJA), and fall (SON) months, respectively.". Keep in mind this number will not match that given on Line 473 in the original manuscript as not all TOLNet observations pass co-location spatiotemporal thresholds.

*Line 176: Add the positions of the ozonesonde stations on the TOLNET map (Fig.1). What is the seasonal distribution of the soundings?*

Figure 1 was provided to inform the readers about the home stations for each of the lidar systems in TOLNet. The spatial locations of the lidar systems, and ozonesondes, are provided in Table 1 of the original manuscript (now Table 2 of the revised manuscript). In order to remind the readers to find this location information the following text was added to the manuscript: "In order to have a direct comparison of the validation using ozonesonde and TOLNet, we use ozonesondes which were nearly directly spatially and temporally co-located with lidar systems as shown in the location information provided in Table 2.".

Text was added to the revised manuscript to describe the seasonal distribution of the ozonesonde data: "The seasonal distribution of these ozonesondes were: 2, 2, 39, and 8 for the winter, spring, summer, and fall months, respectively.".

*Line 236: In equation 3, I guess Xc is the convolved observations using the satellite AK.*

That is correct. To clarify this, we have updated this sentence to read: "The satellite retrievals were compared to raw observations and when convolved ($X_c$) with the averaging kernel ($\textbf{AK}$) and a priori information from each retrieval using Eq. (3)".

*Line 245: I guess the " known TOLNet O3 profile" is the black curve in Fig. 4. Please be more specific here.*

This is now described in the revised manuscript as: "…replaced with a known TOLNet $O_3$ profile (black lines in Fig. 4).".

*Line 285: Once it has been stated that UV-only has limited information below 15 km, I suggest changing the way the end of this sentence is written to focus on the comparison with IR-only:*

*«are much improved (8-10 km) compared to UV-only profiles below 15 km asl. » by*

*«are improved (8-10 km) compared to IR-only above 12 km and below 8 km ».*

We agree with the review and this sentence now reads as: "When combining UV and IR information vertical resolutions of the retrievals are improved (8-10 km) compared to IR-only above 12 km and below 8 km.".

*Line 306: Fig. 4 is a very nice figure and a useful addition to Mettig et al. study. I disagree with the statement « demonstrate the capability of the UV-only, IR-only, and UV+IR retrievals to replicate tropospheric and lowermost tropospheric O3 during a PBL pollution event ». None of the configurations is able to reproduce the ozone enhancement in the lowermost troposphere. It is not so surprising considering the low value of the AK below 2km. It is better to emphasize the very good results obtained for the stratospheric intrusion case for UV+IR, where TOPAZ avoids the downward propagation of the upper tropospheric enhancement compare to IR only.*

We apologize for the confusion the wording of this sentence caused. The purpose of this sentence was simply to introduce/describe Fig. 4. However, as written in the original manuscript, it reads more as the TOPAS retrievals using all wavelengths were able to replicate $O_3$ profiles during pollution and stratospheric intrusions cases. As stated by the reviewer, this is not the case for the PBL pollution event. We have edited this sentence to read: "Figure 4b and 4c demonstrate whether the UV-only, IR-only, and UV+IR retrievals were able to replicate tropospheric and lowermost tropospheric $O_3$ during a PBL pollution event and a stratospheric intrusion, respectively.". The rest of this section in the original manuscript describes the inability of the retrievals to capture the enhanced $O_3$ values in the PBL and ability of the UV+IR products to capture tropospheric $O_3$ during a stratospheric intrusion event.

*Fig. 4. Considering the very high value of this figure, I suggest to add the 6-12 km NMB in panel c to discuss the ability of the 3 configurations to reproduce the upper tropospheric enhancement.*

This is a good point. We have added the NMB values for the mid- to upper-troposphere (4-12 km) for the stratospheric intrusion case study shown in Fig. 4c. We also added text to the revised manuscript to describe this evaluation: "In the mid- to upper-troposphere (4-12 km), UV+IR retrievals had the least high bias (NMB) of 11.3% while IR-only (12.8%) and UV-only (15.9%) retrievals had larger high biases. Compared to the a priori, true lidar profiles convolved with all three retrieval AKs compared much more accurately emphasizing the ability of these retrievals to capture enhanced mid- to -upper tropospheric $O_3$ enhancements.".

*Line 314: Yes I agree with this last statement. Why is this result different from the Cuesta et al. comparison between chemical transport model and combined analysis of GOME-2 and IASI showing a reasonable agreement for ozone enhancement below 3 km? This is worth to be discussed in section 4.*

Thank you for pointing out these differences. The following text has been added to Sect. 4 of the revised manuscript: "Applying different combinations of UV+IR joint wavelength retrievals (e.g., GOME-2+IASI) also displays improvements compared to UV-only products in the troposphere similar to that determined in this study (e.g., Cuesta et al., 2013, 2018). Cuesta et al. (2013, 2018) demonstrated how GOME-2+IASI retrievals show high accuracy compared to ozonesondes in the lowermost troposphere and displays a clear capability to capture PBL $O_3$ enhancements. This differs from the results of this study which suggest that TROPOMI+CrIS UV+IR joint wavelength retrievals still struggle to reproduce large PBL $O_3$ enhancements due to limited lowermost tropospheric sensitivity. The reasons why GOME-2+IASI displays the remarkable capability to retrieve lowermost tropospheric enhancements compared to the results from TROPOMI+CrIS is not immediately apparent. There are differences in the retrieval algorithms, a priori input data sets, and the spectral resolutions of the UV and IR sensors applied. Comparing our results to Cuesta et al. (2013) shows that DOFs are higher in the troposphere and in the 0-2 km agl column (>33% higher) in GOME-2+IASI retrievals compared to TROPOMI+CrIS which would explain some of the differences in capabilities to retrieve lowermost tropospheric $O_3$ enhancements.".

*Table 2. Why are the numbers of observations different in the different vertical layers? Altitude range of the lidar profiles? Clouds?*

We provided this explanation in response to a comment above. Due to limitations of lidar retrievals to accurately retrieve $O_3$ values in high cloud and/or aerosol conditions, and when mid-day solar background becomes too large and saturates the lidar signal, the numbers in Table 2 will differ for each vertical level.

*Line 370: I would suggest discussing all the results in Table 2, including RMSE and bias, in this paragraph instead of mixing them with other topics of Section 4, which should be limited to a general discussion and comparisons with previous works.*

The paragraph discussing RMSE in Sect. 4 in the original manuscript has now been moved into Sect. 3.3.1 of the revised manuscript where Table 2 (now Table 3 in the updated manuscript) statistics are discussed.

*Fig. 5 and Table 2. I do not understand the 89-number of observations in Fig. 5 caption while Table 2 shows up to 172 colocations. Better to have IR-only and UV+IR on the same page in Table 2.*

The total number of satellite/TOLNet profile co-locations using the spatiotemporal co-location criteria of 2.5 hours and 30 km resulted in 89 co-locations. The TOPAS retrieval provides data at 1 km, thus more than one satellite/TOLNet co-location for each of the 2 km bins is possible for each of the 89 total profile co-locations. This is why the numbers in Table 2 for each 2 km bin are larger than 89.

The revised manuscript has been updated so all tables are on the same page.

*Fig.6 and Table 3. Again I do not understand the 26-number in the Fig. 6 caption while 50 soundings are considered in Table 3.*

The answer for this comment is the same for ozonesondes as described above for satellite/TOLNet co-locations.

*Line 375-385: It is a pity that the differences with the TOLneT comparison are not highlighted. This paragraph sounds very positive while the differences with the ozonesonde-raw are significant below 4 km. The improvement using IR+UV instead of IR-only is not obvious anymore for this subset (NMB in Table 3). Is it because the ozonesonde profiles include several cases with lowermost tropospheric enhancement as shown in the sensitivity study in Fig.4b ?*

Text was added to the revised manuscript to better describe the comparison of TOPAS products with ozonesondes and TOLNet data. In the paragraph describing the evaluation with Ozonesonde-AK we added the following sentence: "The evaluation of TOPAS retrievals with Ozonesonde-raw differs from results using TOLNet-raw primarily below 4 km where ozonesonde observed large $O_3$ enhancements in the lowermost troposphere which were not evident in the TOLNet data.". The final paragraph of Sect. 3.3.1 discussed the similarities and differences between the validation of TOPAS retrievals with TOLNet-AK and Ozonesonde-AK. It has been updated slightly in the revised manuscript to better explain this comparison: "In the troposphere, UV-only retrievals were consistently biased high compared to Ozonesonde-AK data (see Fig. 6a, b; Table 4). This

systematic high bias is consistent with the validation using TOLNet-AK observations. IR-only $O_3$ profiles compare very well to Ozonesonde-AK data with NMB values <3% throughout the troposphere. This outperforms the IR-only profiles when compared to TOLNet-AK data which displayed a low bias aloft. Finally, the UV+IR retrievals have minimal bias below 10 km asl with NMB values <10% when compared with TOLNet-AK observations; however, when compared with Ozonesonde-AK data the UV+IR retrievals had a noticeable high bias above 9 km. The overall validation of the three satellite $O_3$ profile retrievals using Ozonesonde-AK was generally consistent compared to when using TOLNet-AK. It is important to note that TOLNet and ozonesonde validation statistics are generally consistent given the fact that ozonesondes are a highly-accurate and commonly-applied satellite validation data source. This suggests that TOLNet is a sufficient validation data source of tropospheric $O_3$ profile retrievals from satellites. Given that TOLNet is able to accurately validate satellite-derived $O_3$ profiles, and the focus of this work is on the demonstration of TOLNet for validating satellite retrievals, the rest of this study focuses on the validation using the lidar network observations.".

*Line 385. As mentioned for Table 2 it is good also to include the RMSE and bias analysis of Table 3 in this paragraph. By the way why is IR-only RMSE smaller than UV+IR RMSE? This should be discussed.*

We agree with the reviewer and the following paragraph has been added to the revised manuscript: "The RMSE values in Table 4 represent the random errors in the daily TOPAS $O_3$ profile retrievals when validated with Ozonesonde-AK observations. All three TOPAS retrievals had lower random errors compared to the a priori profiles; however, random errors still remained elevated in most instances except for the IR-only retrievals. UV-only retrievals had unresolved errors ~50% less compared to the a priori (13.9 ppb). IR-only retrievals displayed the least unresolved errors of all three retrievals with average RMSE values of 6.1 ppb which is ~80% less compared to the a priori. The combined UV+IR profiles had average RMSE values of 11.4 ppb, ~60% less compared to the a priori, throughout the troposphere. Given that unresolved errors of daily profiles on average still remain large (>10 ppb) for retrievals using UV wavelengths (UV, UV+IR), the accuracy of these satellite products still suffer due to the limited sensitivity of spaceborne sensors to tropospheric $O_3$. On the contrary, NMB and RMSE values for IR-only retrievals when compared to Ozonesonde-AK observations were low suggesting this product had some skill in capturing the daily vertical distributions of $O_3$ in the troposphere during this validation. This increased tropospheric sensitivity in IR-only profiles, and when combining UV and IR wavelengths, allows these retrievals to deviate from a biased a priori profiles which improves the ability of this retrieval to capture daily $O_3$ vertical profile distribution variability in the troposphere which is agreement with many recent studies (e.g., Landgraf and Hasekamp, 2007; Worden et al., 2007b; Cuesta et al., 2013, 2018; Costantino et al., 2017; Colombi et al., 2021; Malina et al., 2022; Mettig et al., 2022).".

*Line 428: This sentence is relevant for the validation of the future TEMPO-GEO mission. It is not mandatory for the analysis of the satellite data of this paper where lidar and ozonesonde observations are equally relevant. It is a pity that the ozonesonde data are not included in Fig. 7. The latter ozone vertical distributions are indeed different and complementary from those corresponding to the TOLNET observations (see the comparison between Fig. 5 and 6).*

We have updated this sentence to now read: "A major advantage of using TOLNet for validation of satellite $O_3$ profile retrievals is the ability to make accurate, high temporal and vertical resolution observations at different vertical levels of the troposphere.".

The main objective of this manuscript was to demonstrate the capability of using TOLNet to validate satellite $O_3$ profile retrievals. Therefore, Fig. 7 in the revised manuscript still only applies TOLNet observations. However, based on the reviewers comment below, we now add ozonesonde data with TOLNet to validate seasonal TOPAS retrievals (Fig. S2 in the revised manuscript) to increase the number of seasonal co-locations.

*Line 441: The results of RMSE and slopes in the 4-6 km are not much better than those in the layers 0-4 km even for the IR-only and UV+IR while the DOF are significantly larger than below 4 km, e.g. the slopes in the layer 4-6 km in Fig. 7 are < 0.5 in the Table 2 and Fig. 7. The reason for this limited improvement of IR or IR+UV could be discussed in this section, instead of focusing again on the limitation of UV-only configuration. The latter is already very well demonstrated by the results presented in p. 13 to p.18.*

We now provide more quantitative information for the comparison in these two layers: "Between 2-4 km the UV+IR (NMB of 5.8%, RMSE of 11.7 ppb, slope of 0.46) and especially IR-only (NMB of 4.9%, RMSE of 6.5 ppb, slope of 0.54) retrievals outperform UV-only retrievals (NMB of 18.0%, RMSE of 14.6 ppb, slope of 0.14) due to the enhanced sensitivity provided by the IR wavelengths. The UV+IR and IR-only retrievals have better linear regression slopes compared to the UV-only product (UV-only results have similar slopes as the a priori profile below 6 km) due to the ability to deviate further from the a priori profile shape. In the vertical layer between 4-6 km, similar to the layer between 2-4 km, the UV+IR (NMB of 6.2%, RMSE of 12.3 ppb, slope of 0.45) and in particular the IR-only (NMB of 2.4%, RMSE of 7.5 ppb, slope of 0.62) retrievals outperform UV-only (NMB of 20.1%, RMSE of 16.0 ppb, slope of 0.20) retrievals with less bias and RMSE and better linear regression slopes.". The following text has been added following this to discuss the differences in performance between 2-4 km and 4-6 km: "It should be noted that the retrievals without UV wavelengths (IR-only) was the only satellite product with improved statistics (lower NMB and higher slope) at 4-6 km compared to 2-4 km. The vertical level around 4-6 km is where IR-wavelengths have peak sensitivity to $O_3$ in the TOPAS CrIS retrieval, which contributes to this result. However, given that DOFS for $O_3$ profile retrievals are < 1.0 below 12 km agl, no individual 2 km layer evaluated in this study is completely independent from the retrieval performance throughout the troposphere.". It is important to note that each 2 km layer is not independent and is also driven by retrieval performance at all vertical levels throughout the troposphere.

*Line 471-475: The seasonal analysis is indeed a nice contribution of this paper. However the number of limited co-locations being a limitation of the interpretation of the results, once again the use of the ozonesonde data as well as the TOLNET observations would improve the value of such an analysis.*

We agree with the reviewer that including both TOLNet and ozonesonde data to validate the satellite retrievals would increase the number of seasonal co-locations. In the updated manuscript we include a supplemental figure with this validation. The following text has been added to the

revised manuscript: "The focus of this study was to demonstrate the capability of TOLNet data to validate satellite retrievals; however, to improve the number of seasonal co-locations we also used ozonesonde data (Ozonesonde-AK), in addition to lidar measurements, and these results are shown in Fig. S2. Given the performance of the validation was similar when using TOLNet-AK and the combination of TOLNet-AK and Ozonesonde-AK, the main text of the paper focuses on the seasonal validation of TOPAS retrievals using TOLNet-AK data only.".

*Line 504: I disagree with this statement. The IR-only shows better results below 9 km and the UV+IR SON differences in Fig. 8 are often larger than 10%.*

We thank the reviewer for catching this mistake and this paragraph has been updated in the revised manuscript as: "At all altitudes in the troposphere during the fall months the retrievals using IR wavelengths (IR-only and UV+IR) compared the best to observations with NMB values <15%. UV-only retrievals had consistent high biases typically >20%. IR-only profiles had the best overall performance with small biases (within ±10%) below 9 km and a larger negative bias aloft. All three retrievals had smaller RMSE values compared to the a priori of 16 ppb, 10 ppb, and 13 ppb for the UV-only, IR-only, and UV+IR retrievals, respectively. Similar to the summer months, during the fall all three retrievals had noticeably lower random errors compared to the a priori profiles.".

*Line 526. The sentence is not complete*

To address another reviewer's comment this sentence was removed. We have demonstrated this point throughout the manuscript and it is not needed here.

*Line 527-537. The discussion of RMSE values of Table 2 and 3 would be understood if presented in section 3.3.1 where Fig. 5 and 6 and other statistical parameters of Table 2 and 3 are presented. Mixing this RMSE analysis with a general discussion of the value of the paper results and with a comparison with previous work make reading of this paragraph a little bit difficult.*

As mentioned to a previous comment we have moved the RMSE discussion to Sect. 3.3.1 in the revised manuscript.

*Line 567-568: Remove or change the part of the sentence saying "more capable of capturing conditions with air quality impacts such as pollution events " because this paper does not show this paper does not provide strong evidence for this. It is mainly shown that the stratospheric intrusions are better reproduced.*

This sentence has been revised to read: "Retrievals using combinations of wavelengths proved to be more capable of capturing conditions with air quality impacts such as stratospheric intrusions.".

*Line 574: Again remove the end of the sentence saying "during times of PBL-level O3 enhancements" as it is not clearly shown in this paper (see Fig. 4 or Fig. 6).*

This sentence has been removed in the revised manuscript.

**References:**

Malina, E., Bowman, K. W., Kantchev, V., Kuai, L., Kurosu, T. P., Miyazaki, K., Natraj, V., Osterman, G. B., and Thill, M. D.: Joint spectral retrievals of ozone with Suomi NPP CrIS augmented by S5P/TROPOMI, EGUsphere [preprint], https://doi.org/10.5194/egusphere-2022-774, 2022.

Mettig, N., Weber, M., Rozanov, A., Burrows, J. P., Veefkind, P., Thompson, A. M., Stauffer, R. M., Leblanc, T., Ancellet, G., Newchurch, M. J., Kuang, S., Kivi, R., Tully, M. B., Van Malderen, R., Piters, A., Kois, B., Stübi, R., and Skrivankova, P.: Combined UV and IR ozone profile retrieval from TROPOMI and CrIS measurements, Atmos. Meas. Tech., 15, 2955–2978, https://doi.org/10.5194/amt-15-2955-2022, 2022.

---

## Author Response (AR2)

**Final Response to Reviewers**

We thank the reviewers and editor for their final detailed comments on the manuscript. We have addressed these comments as described below. All reviewer/editor comments are presented in italic font while the author responses are displayed in standard font.

*1) In the first two sentences in the abstract you say "applied" but I think you mean "used"*

In accordance with this reviewer's comment, we have replaced "applied" with "used" in the first two sentences of the abstract in the revised manuscript.

*2) Line 46 : The phrase "random bias" appears… I think you mean something else here.*

Thank you for catching this, we meant to say random errors. This has been corrected in the revised manuscript.

*3) Line 268: Most manuscripts just use "A" instead of "AK". Consider using A alone because AK at first glance could mean A (matrix) multiplied by the Jacobian matrix K.*

Good point. In the revised manuscript, all occurrences of "AK" have been replaced by "$A$" or "averaging kernel".

*4) Im not sure how Figure 7 and the corresponding discussion is useful for this paper. I would not expect 2km layers in the retrieved profile to have that much information about ozone variability (hence the reason for all the scatter). Consider coarsening the layers (e.g. 6 km) to be consistent with the DOFS in the tropospheric part of the profile. Alternatively just remove this figure and section as its confusing and you have already made your points about TOLNET / TROPOMI / CRIS comparison in previous sections.*

To address this final Reviewer #2 comment, we substantially revised Sect. 3.3.3 of the manuscript. The original Fig. 7 showing the evaluation of TOPAS retrievals at 2-km vertical layers has been moved to the supplemental information as Fig. S3. Text S1 has been added to the supplemental information in order to describe the results of Fig. S3. The updated Fig. 7 now shows the scatter plots of the TOPAS retrievals compared to TOLNet observations for two separate 6-km vertical layers in the troposphere. The text in Sect. 3.3.3 has been revised to discuss this new figure.